# High body temperature increases gut microbiota-dependent host resistance to influenza A virus and SARS-CoV-2 infection

Minami Nagai[1,12], Miyu Moriyama [1,12], Chiharu Ishii[2,12], Hirotake Mori [3], Hikaru Watanabe[4], Taku Nakahara[4], Takuji Yamada [4,5], Dai Ishikawa[4,6,7], Takamasa Ishikawa[2], Akiyoshi Hirayama [2], Ikuo Kimura [8,9], Akihito Nagahara[6,7], Toshio Naito[3] ✉, Shinji Fukuda [2,4,6,10,11] ✉ & Takeshi Ichinohe [1] ✉

Fever is a common symptom of influenza and coronavirus disease 2019 (COVID-19), yet its physiological role in host resistance to viral infection remains less clear. Here, we demonstrate that exposure of mice to the high ambient temperature of 36 °C increases host resistance to viral pathogens including influenza virus and severe acute respiratory syndrome coronavirus 2 (SARS-CoV-2). High heat-exposed mice increase basal body temperature over 38 °C to enable more bile acids production in a gut microbiota-dependent manner. The gut microbiota-derived deoxycholic acid (DCA) and its plasma membrane-bound receptor Takeda G-protein-coupled receptor 5 (TGR5) signaling increase host resistance to influenza virus infection by suppressing virus replication and neutrophil-dependent tissue damage. Furthermore, the DCA and its nuclear farnesoid X receptor (FXR) agonist protect Syrian hamsters from lethal SARS-CoV-2 infection. Moreover, we demonstrate that certain bile acids are reduced in the plasma of COVID-19 patients who develop moderate I/II disease compared with the minor severity of illness group. These findings implicate a mechanism by which virus-induced high fever increases host resistance to influenza virus and SARS-CoV-2 in a gut microbiota-dependent manner.

Respiratory infectious diseases, such as influenza and coronavirus disease 2019 (COVID-19), cause substantial morbidity and mortality. These infectious diseases mostly affect older adults[1,2]. Increased susceptibility to influenza virus infection in older humans could be explained by the fact that older human monocytes have impaired signaling to induce type I interferons (IFNs) in response to influenza virus infection[3,4]. However, the role of other age-related changes of host factors in susceptibility to influenza virus infection remains

[1]Division of Viral Infection, Department of Infectious Disease Control, International Research Center for Infectious Diseases, Institute of Medical Science, The University of Tokyo, Tokyo, Japan. [2]Institute for Advanced Biosciences, Keio University, Yamagata, Japan. [3]Department of General Medicine, Juntendo University Faculty of Medicine, Tokyo, Japan. [4]Metagen Therapeutics, Inc., Yamagata, Japan. [5]Department of Life Science and Technology, Tokyo Institute of Technology, Tokyo, Japan. [6]Laboratory for Regenerative Microbiology, Juntendo University Graduate School of Medicine, Tokyo, Japan. [7]Department of Gastroenterology, Juntendo University Faculty of Medicine, Tokyo, Japan. [8]Laboratory of Molecular Neurobiology, Graduate School of Biostudies, Kyoto University, Kyoto, Japan. [9]Department of Applied Biological Science, Graduate School of Agriculture, Tokyo University of Agriculture and Technology, Tokyo, Japan. [10]Gut Environmental Design Group, Kanagawa Institute of Industrial Science and Technology, Kanagawa, Japan. [11]Transborder Medical Research Center, University of Tsukuba, Ibaraki, Japan. [12]These authors contributed equally: Minami Nagai, Miyu Moriyama, Chiharu Ishii.
✉e-mail: naito@juntendo.ac.jp; sfukuda@sfc.keio.ac.jp; ichinohe@ims.u-tokyo.ac.jp

unclear. With respect to age-related changes of host factors, several studies demonstrated that the composition of gut microbiota changes with age in both humans and animals[5–7]. In addition, mean body temperature in humans decrease with age[8]. Although it has increasingly become evident that gut microbiota and their metabolites are important for protection against influenza virus infection[9–12], the effects of body temperature on host defense to influenza virus infection are largely unknown.

In addition to age-related changes of host factors, environmental parameters and eating behavior could affect host susceptibility to influenza virus infection[13–16]. A recent study showed that mice housed in 10 or 20% relative humidity succumbed to influenza virus infection more rapidly than those housed in 50% relative humidity[13]. Further, ketogenic diet-fed mice had improved survival relative to mice on a normal chow diet after influenza virus challenge[14]. In vitro studies using rhinovirus and influenza virus showed that warm temperature restricts viral replication through type I IFN-dependent and -independent mechanisms[17–19]. In addition, both humidity and temperature affect the frequency of influenza virus transmission among guinea pigs[20,21]. We previously demonstrated that exposure of mice to the high ambient temperature of 36 °C impairs virus-specific CD8+ T cell responses and antibody production after intranasal challenge with a sublethal dose (30 pfu) of influenza virus[11]. In contrast, the effects of outside temperature on host resistance to lethal influenza virus challenge remain unknown. The highest temperature during summer reaches 36 °C in Tokyo. In contrast, the average minimum temperature in January goes down to around 4 °C in Tokyo. Thus, here we explore the effects of outside temperatures such as 4 °C or 36 °C on host resistance to influenza virus infection.

In this study, we demonstrate that exposure of mice to the high ambient temperature of 36 °C increases body temperature and host resistance to influenza virus or SARS-CoV-2 infection. Moreover, we demonstrate that high body temperature-dependent activation of gut microbiota increases the levels of bile acids in the serum and intestine,

and suppresses virus replication and detrimental inflammatory responses following influenza virus and SARS-CoV-2 infection.

## Results

### Temperature affects severity of influenza

In mice infected with influenza virus, reduction of body temperature became apparent starting around day 4 p.i. (Supplementary Fig. 1a). Consistent with the reduction of body temperature, the virus-infected mice remained huddled together probably to avoid reducing their body temperature (Supplementary Fig. 1b). These observations prompted us to explore the functional role of body temperature in resistance to influenza virus infection. To this end, we kept mice at 4, 22, or 36 °C for 7 days before influenza virus infection[11]. We have previously confirmed that cold or high-heat exposure of naïve mice was generally well tolerated[11]. Cold-exposed naïve mice exhibited significant decrease in body temperature compared with room temperature-exposed group, whereas high-heat exposure of naïve mice significantly increased body temperatures ranged from 38.4 °C to 39.1 °C (Fig. 1a). To examine the role of body temperature in protection against influenza virus infection, cold-, room temperature-, or high heat-exposed mice were infected intranasally with a mouse-adapted influenza A virus strain A/Puerto Rico/8/1934 (PR8) and kept at 4, 22, or 36 °C, respectively, for the entire duration of the experiments. After influenza virus infection, cold-exposed mice succumbed to disease with severe hypothermia faster than room temperature-exposed group, whereas high heat-exposed mice were highly resistant to influenza virus infection at three challenge doses (Fig. 1b, c and Supplementary Fig. 2). Consistent with our previous report[11], the virus titers were significantly reduced in the lung of high heat-exposed mice compared with cold- or room temperature-exposed groups at early time points (Fig. 1d). Similar results were observed when mice were challenged with influenza virus intratracheally (Supplementary Fig. 3) or a human isolate of the 2009 pandemic influenza A virus strain A/Narita/1/2009 (pdm09) (Supplementary Fig. 4). These protective

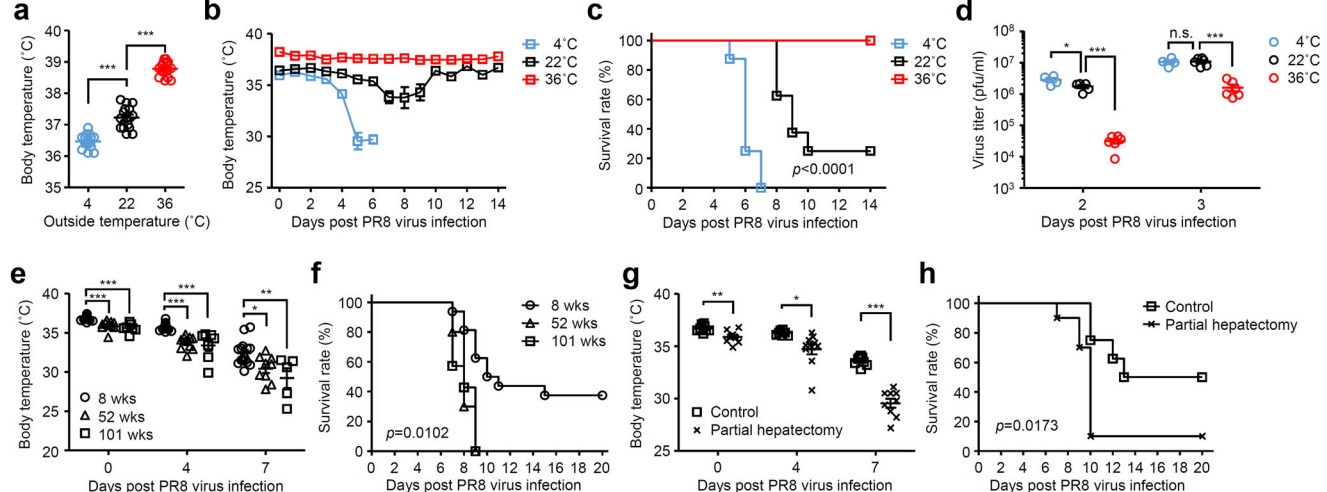

**Fig. 1 | Body temperature affects severity of influenza virus infection.** Mice were kept at 4, 22, or 36 °C for 7 days before influenza virus infection and throughout infection. **a,** Body temperature of naïve mice kept at 4, 22, or 36 °C were measured (4 °C, n = 16 mice; 22 °C, n = 15 mice; 36 °C, n = 15 mice). **b–d** Mice kept at 4, 22, or 36 °C were infected intranasally with 1000 pfu of influenza virus. Body temperatures (**b**; 4 °C, n = 8 mice; 22 °C, n = 8 mice; 36 °C, n = 9 mice), mortality (**c**; 4 °C, n = 8 mice; 22 °C, n = 8 mice; 36 °C, n = 9 mice), and virus titer in the lung wash (**d**; 4 °C, n = 4 mice; 22 °C, n = 5–6 mice; 36 °C, n = 6 mice) were measured on indicated days after challenge. **e, f,** Eight-, 52-, 101-weeks-old mice kept at 22 °C were infected intranasally with 1000 pfu of influenza virus. Body temperatures (**e**) and mortality (**f**) were measured on indicated days after challenge (8 weeks, n = 16 mice; 52 weeks,

n = 10 mice; 101 weeks, n = 7 mice). **g, h** Partially hepatectomized or control mice kept at 22 °C were infected intranasally with 1000 pfu of influenza virus. Body temperatures (**g**) and mortality (**h**) were measured on indicated days after challenge (control, n = 8 mice; partial hepatectomy, n = 10 mice). Each symbols indicate individual values (**a, d, e, g**). Data are mean ± s.e.m. Data are pooled from two independent experiments (**a**) or are representative of two independent experiments (**b–h**). Statistical significance was analyzed by two-way analysis of variance (ANOVA) (**a, d, e**), two-tailed unpaired Student's t test (**g**), or two-sided log-rank (Mantel-Cox) test (**c, f, h**). *P < 0.05, **P < 0.01, ***P < 0.001, n.s., not significant (**g**; 0 day p.i., p = 0.0023; 4 days p.i., p = 0.0139; 7 days p.i., p = 0.00000044).

effects of high heat-exposed mice against influenza virus infections were dependent on the virus challenge dose (Fig. 1c and Supplementary Figs. 2 and 4).

We next examined the effects of outside temperature on host resistance to SARS-CoV-2, another respiratory virus, infection. Recently, it has been reported that Syrian hamsters are a useful small animal model for SARS-CoV-2 infection[22]. Consistent with a previous report[22], hamsters kept at 22 °C were resistant to infection with an original SARS-CoV-2 strain bearing aspartic acid at position 614 of spike (S) protein (S-614D) (Supplementary Fig. 5a–c). In contrast, cold-exposed hamsters succumbed to diseases with severe hypothermia despite having viral loads similar to room temperature-exposed group (Supplementary Fig. 5). Similarly, cold-exposed K18-hACE2 transgenic mice succumbed to disease with severe hypothermia faster than room temperature-exposed group, whereas the high heat-exposed mice were highly resistant to lethal SARS-CoV-2 infection (Supplementary Fig. 6). Next, we wished to examine the effects of outside temperature on host resistance to non-respiratory viral pathogens such as severe fever with thrombocytopenia syndrome phlebovirus (SFTSV), Zika virus, or encephalomyocarditis virus (EMCV). After infection with SFTSV or Zika virus intravenously, cold-exposed mice succumbed to disease, whereas room temperature-exposed mice were resistant to infection (Supplementary Figs. 7 and 8). In addition, high heat-exposed mice improved survival relative to cold- or room temperature-exposed mice after intraperitoneal EMCV challenge (Supplementary Fig. 9).

Next, we wished to setup an experimental condition in which sex-matched mice kept at room temperature have different body temperatures. Since aged mice had lower body temperature than adult mice[23], we next examined the effects of age in influenza virus-induced mortality. Consistent with published reports[23,24], aged mice had lower body temperature than adult mice and were more susceptible to influenza virus infection (Fig. 1e, f), whereas high-heat exposure of the aged mice had improved survival relative to control mice (Supplementary Fig. 10). Next, we wished to setup an experimental condition

in which age- and sex-matched mice have different body temperatures. Since liver plays a key role in thermogenesis and thermoregulation[25], we next examined the effects of partial hepatectomy in influenza virus-induced mortality. Notably, decrease in basal body temperature of room temperature-exposed adult mice by partial hepatectomy exacerbated influenza virus-induced mortality (Fig. 1g, h). Together, these data indicated that body temperature significantly affects host resistance to a broad range of viral infections including influenza A virus and SARS-CoV-2.

## High fever increases gut microbial metabolism

Next, we wished to determine the threshold of body temperature required for protection against influenza virus infection. To this end, we kept mice at 22, 28, 34, or 36 °C before influenza virus infection. While most high heat-exposed naïve mice kept at 36 °C had basal body temperature over 38 °C, an average of basal body temperature of naïve mice kept at 34 °C was 37.2 °C ranged from 37.0 °C to 37.7 °C (Fig. 2a). Although all mice kept at 36 °C survived from lethal influenza virus infection, the majority of the virus-infected mice kept at below 34 °C succumbed to disease at two challenge doses (Fig. 2b and Supplementary Fig. 11a). In addition, all 36 °C-exposed mice kept their body temperature over 38 °C during influenza virus infection (Fig. 2c and Supplementary Fig. 11b). Further, the virus titers were significantly reduced in the lung of high heat-exposed group compared with 34 °C-exposed group at 2 and 3 days p.i. (Fig. 2d). These data indicated that high body temperature over 38 °C may increase host resistance to influenza virus infection.

Recent studies have demonstrated that gut microbiota-derived metabolites protect from influenza virus-induced mortality by enhancing type I IFN signaling or attenuating the neutrophil influx into the lung[9,26]. Thus, we hypothesized that gut microbial activity may be activated by host body temperature over 38 °C[27,28]. To test whether gut microbiota or microbial metabolites are needed to increase host resistance to influenza virus infection in high heat-exposed mice, we

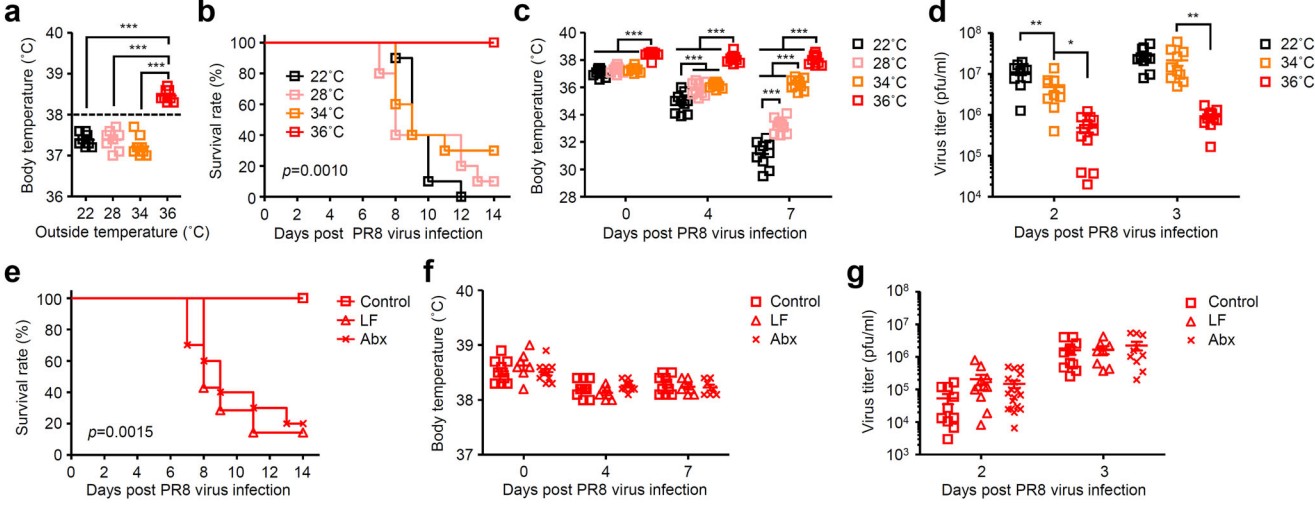

**Fig. 2 | High body temperature increases gut microbiota-dependent host resistance to influenza virus infection.** Mice were kept at 22, 28, 34, or 36 °C for 7 days before influenza virus infection and throughout infection. **a** Body temperature of naïve mice kept at 22, 28, 34, or 36 °C were measured (22 °C, n = 8 mice; 28 °C, n = 8 mice; 34 °C, n = 9 mice; 36 °C, n = 8 mice). The dashed line indicates 38 °C. **b–d** Mice kept at 22, 28, 34, or 36 °C were infected intranasally with 1000 pfu of influenza virus. Mortality (**b**; 22 °C, n = 10 mice; 28 °C, n = 10 mice; 34 °C, n = 10 mice; 36 °C, n = 10 mice), body temperatures (**c**; 22 °C, n = 10 mice; 28 °C, n = 10 mice; 34 °C, n = 10 mice; 36 °C, n = 10 mice), and virus titer in the lung wash (**d**; 22 °C, n = 10 mice; 34 °C, n = 10 mice; 36 °C, n = 11–12 mice) were measured on indicated days after challenge. **e–g** Low fiber (LF)-fed, antibiotics (Abx)-treated, and

control mice kept at 36 °C were infected intranasally with 1000 pfu of influenza virus. Mortality (**e**; control, n = 10 mice; LF-fed, n = 7 mice; Abx-treated, n = 10 mice), body temperatures (**f**; control, n = 10 mice; LF-fed, n = 7 mice; Abx-treated, n = 10 mice), and virus titer in the lung wash (**g**; control, n = 10 mice; LF-fed, n = 8–11 mice; Abx-treated, n = 10–18 mice) were measured on indicated days after challenge. Each symbols indicate individual values (**a, c, d, f, g**). Data are mean ± s.e.m. Data are representative of two independent experiments (**a–f**) or are pooled from two independent experiments (**g**). Statistical significance was analyzed by two-way analysis of variance (ANOVA) (**a, c, d, f, g**) or two-sided log-rank (Mantel-Cox) test (**b, e**). *P < 0.05, **P < 0.01, ***P < 0.001.

infected high heat-exposed low fiber (LF)-fed or antibiotics (Abx)-treated mice with influenza virus. High heat-exposed LF-fed and Abx-treated mice significantly reduced the amounts of DNA isolated from cecal contents compared with high heat-exposed control group (Supplementary Fig. 12a). In addition, high heat-exposed LF-fed and Abx-treated mice changed the bacterial clusters in the cecum compared with high heat-exposed control group (Supplementary Fig. 12b–e). Strikingly, high heat-exposed LF-fed or Abx-treated mice succumbed to disease despite having body temperature over 38 °C similar to high heat-exposed control mice during influenza virus infection (Fig. 2e, f). However, the viral titers in the lung of high heat-exposed LF-fed or Abx-treated mice were similar to that of high heat-exposed control mice at early time points (Fig. 2g). Next, we examined whether gut microbiota or microbial metabolites are needed to increase host resistance to SARS-CoV-2 infection in hamsters. To this end, we infected room temperature-exposed LF-fed or Abx-treated mice with SARS-CoV-2. Interestingly, we found that Abx-treated naïve hamsters kept at 22 °C exhibited significant decrease in basal body temperature compared with control or LF-fed naïve hamsters (Supplementary Fig. 13a). Consistent with a previous report[22], control hamsters kept at 22 °C were resistant to infection with an original SARS-CoV-2 strain bearing aspartic acid at position 614 of spike (S) protein (S-614D) (Supplementary Fig. 13b). In contrast, LF-fed and Abx-treated hamsters succumbed to infection despite having viral loads similar to control hamsters (Supplementary Fig. 13b–d). These data indicated that inability of high heat-exposed LF-fed and Abx-treated mice or room temperature-exposed LF-fed and Abx-treated hamsters to protect against lethal influenza virus or SARS-CoV-2 challenge was not due to their body temperature but may be instead due to reduction of gut microbial fermentation products.

To gain insight into the mechanisms underlying gut microbiota-derived metabolites mediated host resistance to influenza virus infection in high heat-exposed mice, we performed capillary

electrophoresis time-of-flight mass spectrometry (CE-TOFMS), gas chromatography–mass spectrometry (GC-MS), liquid chromatography-tandem mass spectrometry (LC-MS/MS)-based metabolome analysis of cecal contents, serum, and livers of naïve mice kept at 4, 22, or 36 °C for 7 days. The CE-TOFMS-based cecal metabolome profiles were clustered and found that the levels of both primary and secondary bile acids were significantly increased in the high heat-exposed mice or hamsters compared with cold- or room temperature-exposed groups (Supplementary Fig. 14). In contrast, several types of both primary and secondary bile acids were significantly enhanced in the serum and livers of high heat-exposed mice compared with cold- or room temperature-exposed groups (Supplementary Figs. 15 and 16). In addition, both primary and secondary bile acids were significantly reduced in the serum of high heat-exposed LF-fed or Abx-treated mice compared with high heat-exposed control mice (Supplementary Fig. 17).

## Bile acids-TGR5 axis protects from influenza

Thus far, our data indicated that high heat-exposed mice increased host resistance to a broad range of viral infections including influenza A virus and SARS-CoV-2. In addition, high heat-exposed mice may increase gut microbial biotransforming reactions to produce the secondary bile acids. These observations led us to focus on the role of bile acids in resistance to influenza virus infection. We hypothesized that elevated levels of bile acids in the serum of high heat-exposed mice may play an important role in host resistance to viral infection. To address these possibilities, room temperature-exposed mice were given 0.5 mM of primary or secondary bile acids in drinking water for the entire duration of the experiments. After infection with influenza virus, room temperature-exposed mice that had been given cholic acid (CA), deoxycholic acid (DCA), ursodeoxycholic acid (UDCA), or taurine-conjugated deoxycholic acid (TDCA) in drinking water for 7 days prior to infection had improved survival relative to control mice (Fig. 3a, b and Supplementary Fig. 18a, b). The virus titers were

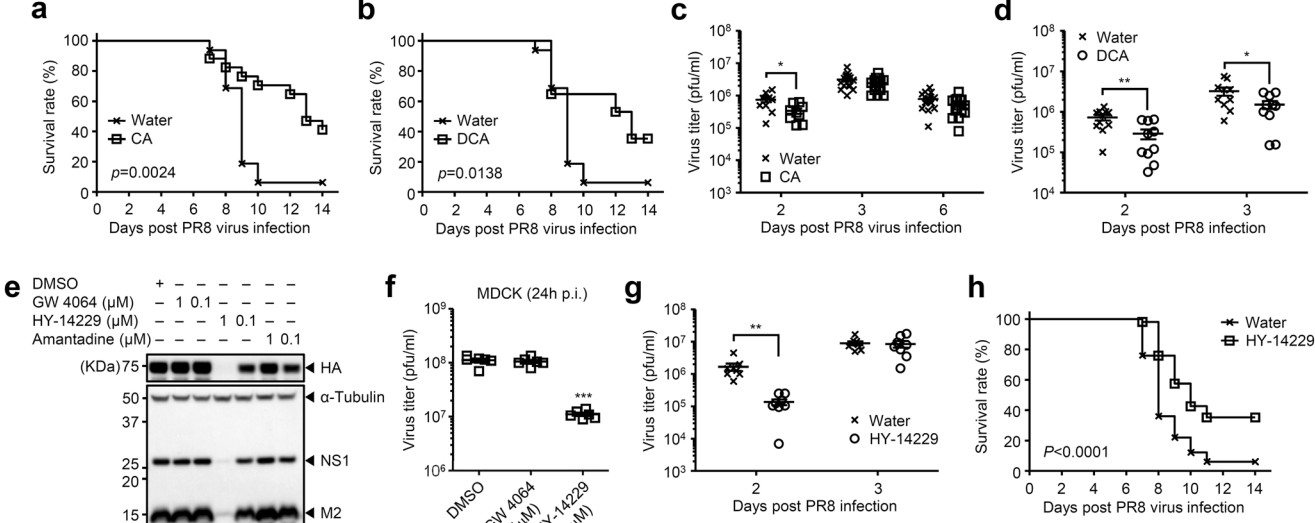

**Fig. 3 | Bile acids-TGR5 axis protects from influenza virus infection. a–d** Room temperature-exposed mice given 0.5 mM of CA (**a**) or DCA (**b**) were infected intranasally with 1000 pfu of influenza virus. Mortality (**a, b**; water, *n* = 16 mice; CA, *n* = 17 mice; DCA, *n* = 17 mice) and virus titer in the lung wash (**c, d**; water, *n* = 10 to 16 mice; CA, *n* = 9 to 16 mice; DCA, *n* = 10 mice) were measured on indicated days after challenge. **e, f** MDCK cells were infected with influenza virus PR8 (an amantadine-resistant strain) in the presence or absence of indicated amounts of GW 4064, HY-14229, or amantadine. Cell lysates were collected at 24 h p.i. and analyzed by immunoblotting with indicated antibodies (**e**). Cell-free supernatants were collected at 24 h p.i. and analyzed for virus titer by standard plaque assay using MDCK cells (**f**; DMSO, *n* = 6; GW 4064, *n* = 6; HY-14229, *n* = 6). **g, h** Room

temperature-exposed mice given 100 μM of HY-14229 were infected intranasally with 1000 pfu of influenza virus. Virus titer in the lung wash (**g**; water, *n* = 8 mice; HY-14229, *n* = 8 mice) and mortality (**h**; water, *n* = 68 mice; HY-14229, *n* = 73 mice) were measured on indicated days after challenge. Each symbols indicate individual values (**c, d, f, g**). Data are mean ± s.e.m. Data are pooled from two (**a–d**) or three (**h**) independent experiments or are representative of two independent experiments (**e–g**). Statistical significance was analyzed by two-way analysis of variance (ANOVA) (**f**), two-tailed unpaired Student's *t* test (**c, d, g**), or two-sided log-rank (Mantel-Cox) test (**a, b, h**). *P < 0.05, **P < 0.01 (**c**; 2 days p.i., *p* = 0.014; **d**; 2 days p.i., *p* = 0.0052; 3 days p.i., *p* = 0.0456; **g**; 2 days p.i., *p* = 0.00321).

significantly suppressed in the lungs of CA-, DCA-, or UDCA-treated mice at 2 days p.i. (Fig. 3c, d and Supplementary Fig. 18c). However, the virus titer in the lungs remained similar in water-fed and CA-treated mice at 3 and 6 days p.i. (Fig. 3c), suggesting that not only suppression of viral load at early time points but also other mechanisms at later time points could explain how treatment of mice with bile acids improve survival after influenza virus infection. In CA-treated mice, the levels of TDCA, glycine-conjugated DCA (GDCA) and glycine-conjugated CA (GCA) but not DCA and UDCA in the serum were significantly elevated compared with water-fed control group (Supplementary Fig. 19), suggesting that not only DCA but also taurine- or glycine-conjugated bile acids in serum play an important role in protection against influenza virus infection.

Since bile acids are digestive surfactants that promote lipid absorption, it is possible that they directly disrupt enveloped viruses including influenza virus and SARS-CoV-2 by destroying membrane integrity[29]. To test this possibility, influenza virus, SARS-CoV-2, or encephalomyocarditis virus (EMCV), a non-enveloped RNA virus, were incubated with various concentrations of DCA at 37 °C for 1 h. Indeed, DCA inactivated the influenza virus and SARS-CoV-2, but not EMCV, at 1.25 mM or higher (Supplementary Fig. 20a–c). However, concentrations of DCA higher than 0.25 mM were toxic to cultured cells (Supplementary Fig. 20d, e).

Several other possible mechanisms could explain how higher concentration of DCA in the serum might increase host resistance to influenza virus infection. First, detergent properties of bile acids might affect viral entry or budding by modulating the integrity of cellular membranes[30,31] (Supplementary Fig. 21a). Second, bile acids activated receptors might regulate virus-induced inflammatory responses[32–34]. First, we examined the direct effects of DCA on influenza virus replication. In the presence of 0.125 mM of DCA, the ratio of influenza virus nucleoprotein (NP)-positive cells was significantly enhanced at 6 h p.i. (Supplementary Fig. 21b, c), which indicated that DCA facilitates influenza virus entry into MDCK cells. However, influenza virus replication was significantly inhibited by DCA at 24 and 48 h p.i. (Supplementary Fig. 21d, e). Next, we wished to determine which bile acid receptors could contribute to inhibition of influenza virus replication. To this end, we infected MDCK cells with influenza virus in the presence or absence of GW 4064 (a FXR agonist) or HY-14229 (a TGR5 agonist) and found that 1 μM of HY-14229 but not GW 4064 inhibited influenza virus replication (Fig. 3e, f and Supplementary Fig. 22). In addition, treatment of the virus-infected cells with 1 μM of HY-14229 by 9 h p.i. inhibited influenza virus replication (Supplementary Fig. 23). Further, HY-14229-treated mice significantly reduced the virus replication at 2 days p.i. and had improved survival relative to water-fed group (Fig. 3g, h).

Next, we tested the second hypothesis that bile acids activated receptors might regulate virus-induced inflammatory responses[32–34]. High heat-exposed mice impaired secretion of mature IL-1β in the lung wash following influenza virus infection (Supplementary Fig. 24)[11]. One of the hallmarks of IL-1 signaling is the recruitment of neutrophils[35,36]. In addition, previous studies have indicated the detrimental role of neutrophils in influenza virus-induced mortality[3,37]. Thus, we first examined the recruitment of neutrophils into the lung of high heat-exposed mice during influenza virus infection. Following influenza virus infection, the levels of pulmonary CXCL1, which is one of the major chemoattractant of neutrophils, in the lung of room temperature-exposed mice became apparent starting around day 3 p.i. and peaking around day 4 p.i. (Fig. 4a). In contrast, the levels of pulmonary CXCL1 were significantly suppressed in the lung of high-heat exposed mice compared with room temperature-exposed group at 4 days p.i. (Fig. 4b). Consistent with this observation, both frequency and number of neutrophils were significantly suppressed in the lung of high-heat exposed mice compared with room temperature-exposed group at 7 days p.i. (Fig. 4c, d). In addition, the levels of CXCL1 and neutrophil recruitment were significantly elevated in the lung of high

heat-exposed LF-fed and Abx-treated mice compared with high heat-exposed control group (Supplementary Fig. 25). We next examined whether bile acids or their receptors inhibit influenza virus-induced CXCL1 production. Treatment of bone marrow-derived macrophages with 125 μM of DCA (LDH release <1%) but not CA significantly inhibited not only IL-1β, an important mediator of CXCL1 production (Supplementary Fig. 26)[38], but also CXCL1 production from influenza virus-infected bone marrow-derived macrophages (Fig. 4e–g). Consistent with this observation, room temperature-exposed mice that had been given DCA in drinking water suppressed neutrophil recruitment into the lung tissue compared with room temperature-exposed water-fed group (Fig. 4h). In addition, HY-14229 but not GW 4064 inhibited influenza virus-induced IL-1β and CXCL1 production from bone marrow-derived macrophages (Fig. 4i, j). Further, depletion of neutrophils or neutrophil extracellular traps digestion in vivo resulted in prolonged survival of room temperature-exposed mice after influenza virus infection (Fig. 4k, l), consistent with previous reports[3,26,39]. Together, these data suggested that elevated levels of secondary bile acids in the serum of high heat-exposed mice may increase host resistance to viral infection by suppressing influenza virus replication and detrimental inflammatory responses.

## Role of bile acids in COVID-19

We next examined inhibitory effect of bile acids against SARS-CoV-2 infection. To this end, we infected VeroE6/TMPRSS2 cells with SARS-CoV-2 in the presence or absence of various bile acids and found that DCA and UDCA inhibited the virus protein synthesis (Fig. 5a). Consistent with this observation, SARS-CoV-2 replication was significantly inhibited by DCA at 48 h p.i. (Fig. 5b). In addition, DCA-treated hamsters had improved survival relative to water-fed group following SARS-CoV-2 B.1.1.7 (alpha) variant infection without affecting the virus titer at 3 days p.i. (Fig. 5c and Supplementary Fig. 27a, b). Although CA did not inhibit SARS-CoV-2 replication in vitro (Fig. 5a), CA-treated hamsters had improved survival relative to water-fed group after infection with SARS-CoV-2 B.1.1.7 (alpha) variant (Fig. 5d and Supplementary Fig. 27c). Cold-exposed hamsters, which exacerbated SARS-CoV-2-induced mortality (Supplementary Fig. 5), exhibited significant decrease in basal body temperature compared with room temperature-exposed group (Supplementary Fig. 28a). In addition, cold-exposed naïve hamsters significantly reduced the levels of glycine-conjugated chenodeoxycholic acid (GCDCA), GDCA, and GCA in the serum compared with room temperature-exposed group (Supplementary Fig. 28b–d). Together, these data suggested that not only DCA but also taurine- or glycine-conjugated bile acids may play an important role in protection against SARS-CoV-2 infection. Interestingly, we found that 100 μM of GW 4064 efficiently blocked SARS-CoV-2 replication in vitro (Fig. 5e and Supplementary Fig. 29). In addition, GW 4064 but not HY 14229 treatment significantly improved survival of hamsters following SARS-CoV-2 B.1.1.7 (alpha) variant infection without affecting the virus titer at early time points (Fig. 5f, g and Supplementary Fig. 30).

Finally, we examined the relationship between the levels of bile acids in the plasma of COVID-19 patients and severity of the disease. To this end, 46 patients who had been admitted to Juntendo University Hospital were stratified into minor or moderate I/II disease groups on the basis of optimal oxygen saturation, clinical symptoms and chest radiographic findings. Demographics and background information for 46 patients are in Supplementary Table 1. The levels of serum amyloid A and fibrinogen, important biomarkers of COVID-19 disease severity, were significantly elevated in the plasma of patients who developed moderate I/II disease compared with minor illness group (Fig. 5h and Supplementary Fig. 31). Importantly, the levels of GCA, taurine-conjugated CA (TCA), taurine-conjugated chenodeoxycholic acid (TCDCA), and GCDCA were significantly reduced in the plasma of COVID-19 patients who developed moderate I/II disease compared

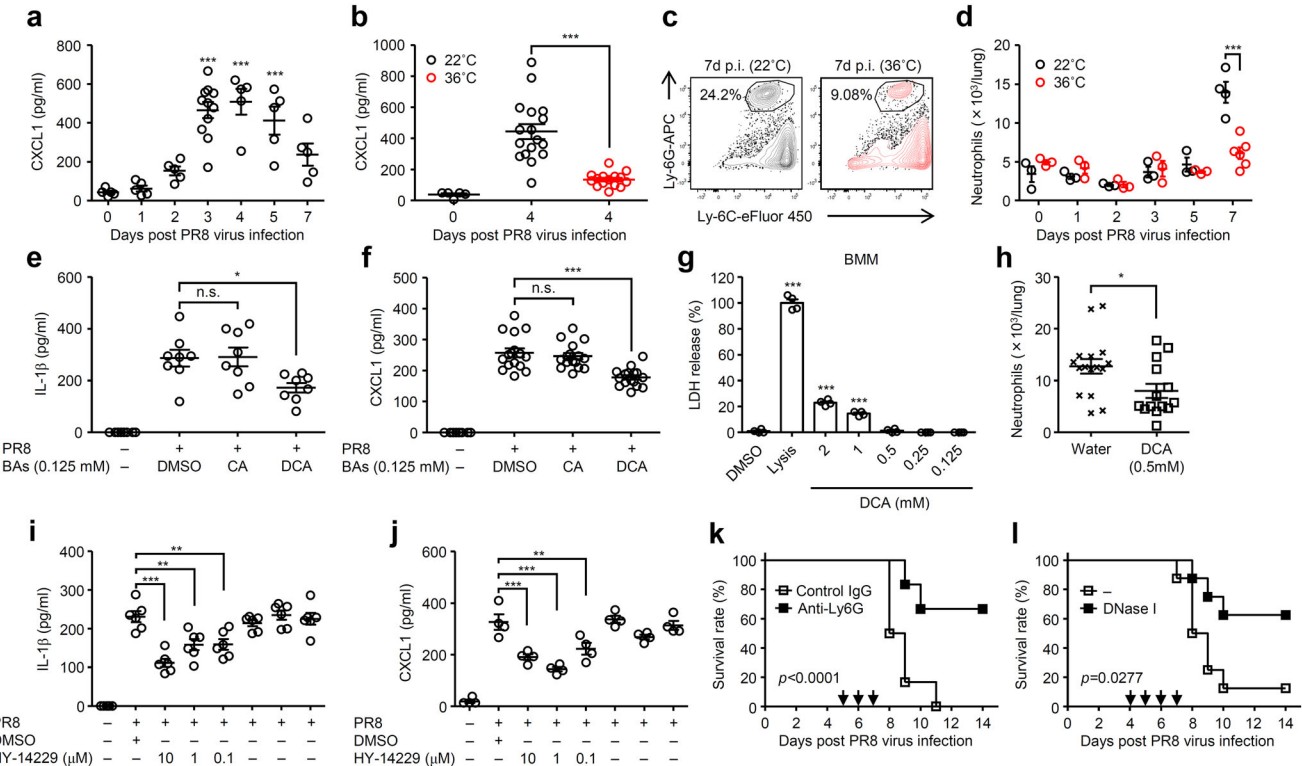

**Fig. 4 | Bile acids-TGR5 axis inhibits inflammatory responses following influenza virus infection. a–d**, Mice kept at 22 or 36 °C were infected intranasally with 1000 pfu of influenza virus. The lung washes were collected at indicated time points and analyzed for CXCL1 by ELISA (**a**, $n = 5$ mice at 0, 1, 2, 4, 5, an 7 days p.i., $n = 11$ mice at 3 days p.i.; **b** $n = 5$ for uninfected, $n = 17$ mice for infected at 22 °C, and $n = 15$ mice for infected at 36 °C). Leukocytes were isolated from the lung at indicated time points. The number of Ly6C⁺ Ly6G⁺ neutrophils were analyzed by flow cytometry (**c**, **d**; $n = 3$ room temperature- or high heat-exposed mice at 0, 1, 2, 3, and 5 days p.i., $n = 4$ room temperature-exposed mice at 7 days p.i., $n = 6$ high heat-exposed mice at 7 days p.i.). **e**, **f** Bone marrow-derived macrophages were infected with influenza virus in the presence or absence of CA or DCA. Cell-free supernatants were collected at 24 h p.i. and analyzed for IL-1β (**e**; $n = 8$ mice) or CXCL1 (**f**; $n = 8$ mice for uninfected, $n = 16$ mice for infected) by ELISA. **g** Uninfected bone marrow-derived macrophages were cultured in the presence or absence of indicated amounts of DCA for 24 h. LDH activity was measured for cytotoxicity ($n = 4$). **h** DCA-treated or control mice kept at 22 °C were infected intranasally with 1000 pfu of influenza virus. Leukocytes were

isolated from the lung at 7 days p.i.. The number of Ly6C⁺ Ly6G⁺ neutrophils were analyzed by flow cytometry ($n = 16$ mice for water-fed, $n = 14$ mice for DCA-treated). **i**, **j** Bone marrow-derived macrophages were infected with influenza virus in the presence or absence of indicated amounts of HY-14229 or GW 4064. Cell-free supernatants were collected at 24 h p.i. and analyzed for IL-1β (**i**; $n = 6$) or CXCL1 (**j**; $n = 4$) by ELISA. **k**, **l** Mice kept at 22 °C were infected intranasally with 1000 pfu of influenza virus. A group of mice was treated with either a neutrophil-depleting antibody to Ly6G (**k**; control IgG, $n = 6$ mice; anti-Ly6G, $n = 6$ mice) or recombinant DNase intraperitoneally (**l**; control, $n = 8$ mice; DNase I, $n = 8$ mice) at indicated time points (arrows). Survival was monitored for 14 days. Each symbols indicate individual values (**a**, **b**, **d–j**). Data are mean ± s.e.m. Data are representative of two independent experiments (**a**, **c–e**, **g**, **i–l**) or are pooled from two independent experiments (**b**, **f**, **h**). Statistical significance was analyzed by two-way analysis of variance (ANOVA) (**a**, **b**, **e–g**, **i**, **j**), two-tailed unpaired Student's $t$ test (**d**, **h**), or two-sided log-rank (Mantel-Cox) test (**k**, **l**). *$P < 0.05$, **$P < 0.01$, ***$P < 0.001$, n.s., not significant (**d**; 7 days p.i., $p = 0.00053$; **h**; $p = 0.0246$).

with minor illness group (Fig. 5i and Supplementary Fig. 32). In addition, the levels of the GCA, TCA, TCDCA, and GCDCA in plasma were significantly and negatively correlated with the levels of fibrinogen in plasma but not patients' age (Supplementary Figs. 33 and 34). Treatment of VeroE6/TMPRSS2 cells with 2 mM of GCA (LDH release <1%) efficiently inhibited the viral protein synthesis and replication compared with TCA, TCDCA, or GCDCA (Supplementary Fig. 35). Thus, we next examined the protective effect GCA against SARS-CoV-2 infection in hamsters. GCA-treated hamsters had improved survival relative to water-fed group following SARS-CoV-2 B.1.1.7 (alpha) variant infection without affecting the virus titer at 3 days p.i. (Fig. 5j, k and Supplementary Fig. 36). Taken together, our data show that body temperature-dependent activation of bile acid metabolism by gut microbiota increases host resistance to influenza virus and SARS-CoV-2 infection by suppressing virus replication and detrimental inflammatory responses through bile acid receptors.

## Discussion

Our findings here have identified an unappreciated link between body temperature, gut microbial activity, and host resistance to viral

infection. Specifically, high heat-exposed mice increased their basal body temperature over 38 °C, which may stimulate gut microbial activity[40]. Indeed, both primary and secondary bile acids were significantly enhanced in the serum and livers of high heat-exposed mice. Increased mortality in partially hepatectomized mice following influenza virus infection may be explained by the fact that primary bile acids are synthesized from cholesterol exclusively in the liver and further metabolized by the gut microbiota into secondary bile acids[41]. Because secondary bile acids are produced by microbial biotransforming reactions in the gut[42], high heat-exposed LF-fed and Abx-treated mice significantly reduced the secondary bile acids in the serum and succumbed to influenza virus infection. We have also shown that DCA and HY-14229 (a TGR5 agonist) inhibit influenza virus replication and inflammatory responses both in vitro and in vivo.

It has increasingly become evident that gut microbiota promotes type I IFNs-dependent innate antiviral immunity and confers host resistance to influenza virus infection. We and others have previously shown that Abx-treated mice are more susceptible to influenza virus infection[9,11,43,44]. Abt et al. have shown that alveolar macrophages isolated from Abx-treated mice decreased innate antiviral genes

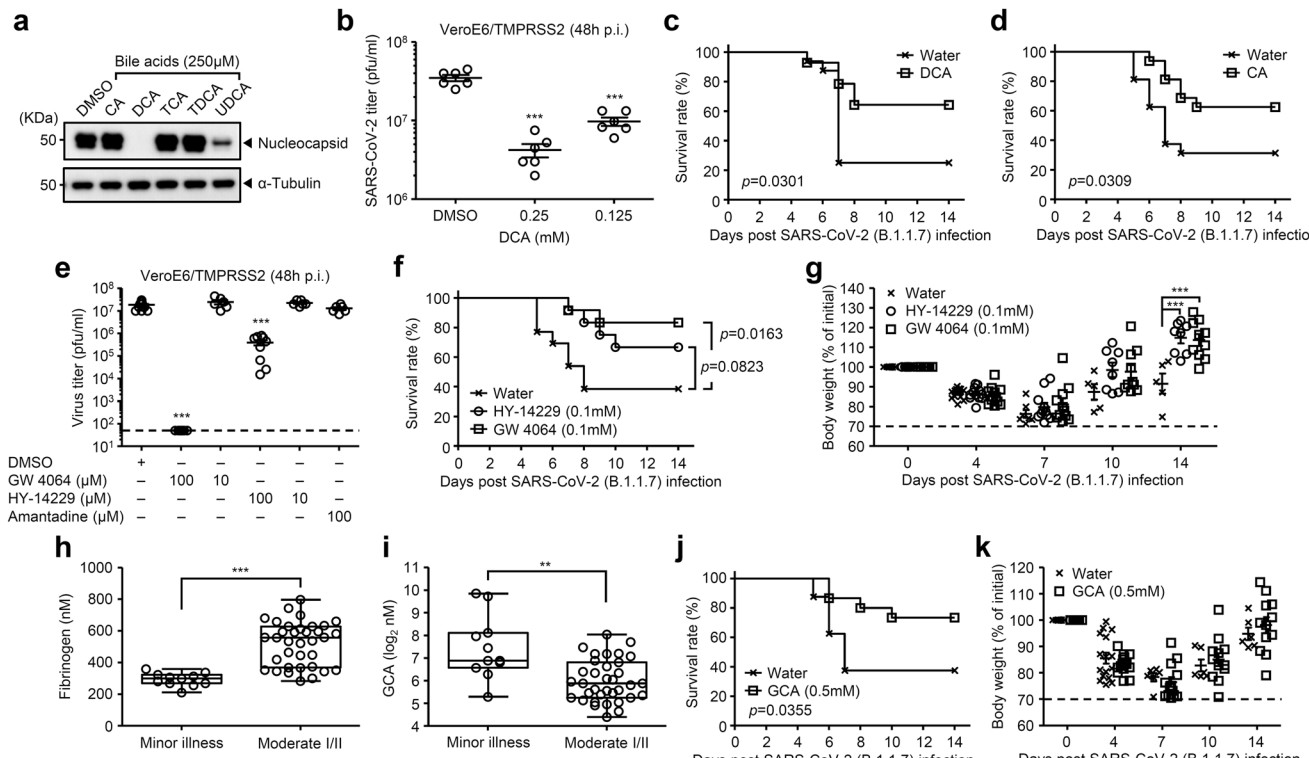

**Fig. 5 | Bile acids-FXR axis protects from SARS-CoV-2 infection. a, b,** VeroE6/ TMPRSS2 cells were infected with original SARS-CoV-2 (S-614D) in the presence or absence of indicated bile acids (**a**) or DCA (**b**). Cell lysates were collected at 24 h p.i. and analyzed by immunoblotting with indicated antibodies (**a**). Cell-free supernatants were collected at 48 h p.i. and analyzed for virus titer by standard plaque assay using VeroE6/TMPRSS2 cells (**b**; *n* = 6). **c, d** DCA- (**c**) or CA-treated (**d**) hamsters kept at 22 °C were infected intranasally with 1.5 × 10⁶ pfu of SARS-CoV-2 B.1.1.7 (alpha) variant. Survival was monitored for 14 days (water, *n* = 16 hamsters; DCA, *n* = 14 hamsters; CA, *n* = 16 hamsters). **e** VeroE6/TMPRSS2 cells were infected with original SARS-CoV-2 (S-614D) in the presence or absence of indicated amounts of GW 4064, HY-14229, or amantadine as a negative control. Cell-free supernatants were collected at 48 h p.i. and analyzed for virus titer by standard plaque assay using VeroE6/TMPRSS2 cells (*n* = 15 tests for DMSO control, *n* = 9 tests for 100 μM of GW 4064 and HY-14229, and *n* = 6 tests for 10 μM of GW 4064 and HY-14229, and 100 μM of amantadine). The lower limit of detection is indicated by the horizontal dashed line. **f, g** HY-14229 (100 μM)-, GW 4064 (100 μM)-treated or control hamsters kept at 22 °C were infected intranasally with 1.5 × 10⁶ pfu of SARS-CoV-2 B.1.1.7

(alpha) variant. Mortality (**f**) and body weight (**g**) were measured on indicated days after challenge (water, *n* = 13 hamsters; HY-14229, *n* = 12 hamsters; GW 4064, *n* = 12 hamsters). The dashed line indicates the limit of endpoint (**g**). **h, i** plasma concentration of fibrinogen (**h**) and GCA (**i**) in minor (*n* = 11 patients) versus moderate I/ II (*n* = 35 patients) groups. Each dot represents a unique patient. **j, k** Control or GCA-treated hamsters kept at 22 °C were infected intranasally with 1.5 × 10⁶ pfu of SARS-CoV-2 B.1.1.7 (alpha) variant. Mortality (**j**) and body wight (**k**) were measured on indicated days after challenge (water, *n* = 16 hamsters; GCA, *n* = 15 hamsters). The dashed line indicates the limit of endpoint (**k**). Each symbols indicate individual values (**b, e, g–i, k**). Data are mean ± s.e.m. Data are representative of two independent experiments (**a, b, f, g, j, k**) or are pooled from two (**c, d**) or three independent experiments (**e**). Statistical significance was analyzed by two-way analysis of variance (ANOVA) (**b, e, g, k**), a two-tailed Mann-Whitney test (**h, i**), or two-sided log-rank (Mantel-Cox) test (**c, d, f, j**). The center line denotes the median value (50th percentile), while the white box contains the 25th to 75th percentiles of dataset. The black whiskers mark the 5th and 95th percentiles (**h, i**). ***P* < 0.001 (**h**; *p* < 0.0001; **i**; *p* = 0.0029).

expression in the lung at 3 days post influenza virus infection. Thus, intranasal administration of Abx-treated mice with poly(I:C) at −1 and 3 days post influenza virus infection improved survival relative to control group[43]. In addition, bone marrow chimera experiments showed that gut microbiota-driven IFN priming in the lungs acts on stromal cells to establish an antiviral state during influenza virus infection[44]. Further, Steed et al. have identified desaminotyrosine as a gut microbiota-derived metabolite that is able to increase interferon-stimulated genes expression in the lungs and host resistance to influenza virus infection[9]. Recent studies indicated that bile acids also increase type I IFN-dependent innate antiviral immunity[45,46]. Hu et al. have indicated that both HSV-1 (DNA virus) and Sendai virus (RNA virus) induces accumulation of intracellular bile acids. The accumulated intracellular bile acids activate SRC kinase through the TGR5-GRK-β-arrestin axis to enable tyrosine phosphorylation of multiple innate antiviral signaling components. Therefore, TGR5 knockout mice were more susceptible to HSV-1 and EMCV infections[45]. Winkler et al. have shown that dysbiosis of gut microbiota increases virus burden in the serum and spleen of Chikungunya virus (CHIKV)-infected mice. However, reconstitution with a single bacterial species, *Clostridium*

*scindens*, or its derived DCA can restore TLR7 and MyD88-dependent type I IFN responses to restrict systemic CHIKV infection[46]. Although the roles of gut microbiota or its derived secondary bile acids in type I IFN-dependent innate antiviral immune responses are well-established, role of body temperature in gut microbial activity and host resistance to influenza virus infection remains unknown. We found that high body temperature over 38 °C significantly increased gut microbiota-derived bile acids in high-heat exposed mice. In addition, we showed that HY-14229 (a TGR5 agonist)-treated room temperature-exposed mice improved survival following influenza virus infection.

Our findings obtained from influenza virus infection applies to SARS-CoV-2 infection. Specifically, high heat-exposed K18-hACE2 mice were highly resistant to SARS-CoV-2 infection compared with room temperature-exposed group (Supplementary Fig. 6). In addition, CA- or DCA-treated hamsters had improved survival relative to water-fed group following SARS-CoV-2 B.1.1.7 (alpha) variant infection (Fig. 5c, d). While GW 4064 (a FXR agonist) inhibited SARS-CoV-2 replication in vitro (Fig. 5e and Supplementary Fig. 29), the GW 4064 treatment significantly improved survival of hamsters following SARS-CoV-2 B.1.1.7 (alpha) variant infection without affecting the virus titer at early

time points (Fig. 5f, g and Supplementary Fig. 30). Importantly, we found that the levels of GCA, TCA, TCDCA, and GCDCA were significantly reduced in the plasma of COVID-19 patients who developed moderate I/II disease compared with minor illness group (Fig. 5i and Supplementary Fig. 32). In addition, GCA-treated hamsters had improved survival relative to water-fed group following SARS-CoV-2 B.1.1.7 (alpha) variant infection (Fig. 5j, k). These observations suggest that secondary bile acids and their receptor signaling may increase host resistance to SARS-CoV-2 infection. Recent studies suggested that gut microbiome dysbiosis during COVID-19 is associated with detrimental host inflammatory responses and severity of the disease[47,48]. Given that the severity of COVID-19 disease is associated with an influx of innate immune cells and inflammatory cytokines[49–51], inhibition of detrimental inflammatory responses by a FXR ligand or probiotic therapy may increase host resistance to SARS-CoV-2 infection.

Our study identifies several possible mechanisms by which a gut microbiota-derived secondary bile acids can increase host resistance to influenza virus or SARS-CoV-2 infection. Several possible mechanisms could explain how bile acids or their receptors signaling increased host resistance to influenza virus or SARS-CoV-2 infection. First, detergent properties of DCA at high concentrations (1.25 mM or higher) directly disrupted influenza virus or SARS-CoV-2 by destroying envelope membrane (Supplementary Fig. 20a, b)[29]. Second, a low concentration of DCA (125 µM) inhibited influenza virus replication by modulating the integrity of cellular membranes[30,31] (Supplementary Fig. 21a). Third, TGR5 may suppress influenza virus replication (Fig. 3e–g) probably by inhibiting NF-κB activity or adhesion molecule expression[52]. Fourth, DCA and TGR5 signaling suppressed neutrophil recruitment into the lung tissue by inhibiting influenza virus-induced IL-1β and CXCL1 production from macrophages (Fig. 4e, f, h–j). These data are consistent with a published report showing that bile acids including DCA inhibit the NLRP3 inflammasome-mediated IL-1β secretion from bone marrow-derived macrophages by stimulating TGR5-cAMP-protein kinase A axis-dependent NLRP3 phosphorylation and ubiquitination[33]. Although bile acids or their receptors signaling inhibited SARS-CoV-2 protein synthesis and replication in vitro (Fig. 5a, b and Supplementary Figs. 29, and 35b, c), they protected hamsters from lethal SARS-CoV-2 infection without affecting the virus titer at early time points (Supplementary Figs. 27b, 30, and 36). Similarly, while the viral titers in the lung of high heat-exposed control mice were similar to that of high heat-exposed LF-fec or Abx-treated mice at 3 days p.i. (Fig. 2g), only high heat-exposed control group survived from lethal influenza virus infection (Fig. 2e). Since excess neutrophil recruitment into the lung exacerbated influenza virus-induced mortality[3,37], these data suggests that inhibition of detrimental inflammatory responses rather than viral replication may be important for gut microbiota-dependent host resistance to influenza virus and SARS-CoV-2 infections in high heat-exposed mice or bile acids-treated hamsters.

In the present study, we demonstrated that high heat-exposed mice kept at 36 °C are highly resistant to influenza virus and SARS-CoV-2 infection. A recent study showed that mice housed in 10 or 20% relative humidity succumbed to influenza virus infection more rapidly than those housed in 50% relative humidity[13]. However, the current study could not maintain the constant humidity in cold-(80% relative humidity), room temperature- (53% relative humidity), and high heat-exposed groups (25% relative humidity). In addition, we previously demonstrated that the high heat-exposed mice decreased their food intake and increased autophagy in lung tissue[11]. Thus, we cannot simply compare the effects of outside and body temperature on host resistance to viral infection. In addition, respiration rates affected by outside temperature could impact the initial viral inoculation[53]. To rule out this possibility, we infected cold-, room temperature-, and high heat-exposed mice with influenza virus intratracheally and found that high heat-exposed mice were highly resistant to influenza virus

infection (Supplementary Fig. 3), suggesting that difference in survival rate of influenza virus-infected cold-, room temperature-, and high heat-exposed mice was not due to the changes of respiration rates or initial viral inoculation. In addition, although treatment of mice or hamsters with 0.5 mM of bile acids or 0.1 mM of HY-14229 or GW 4064 via drinking water was generally well tolerated, it will be important to determine the administration dose and route to maximize the protective effects of bile acids or the agonists against influenza virus or SARS-CoV-2 infection without deleterious impact in vivo. Since high body temperature during fever cause various physiological changes in the body, future studies to dissect other mechanisms by which high body temperature confers host resistance to viral infection will be important extensions of this work.

In summary, our findings substantially expand our understanding of how body temperature increases host resistance to influenza virus and SARS-CoV-2 infection. Our detailed mechanistic analysis demonstrates that high body temperature-dependent activation of gut microbiota metabolism increases the levels of bile acids in the serum and intestine, and suppresses virus replication and detrimental inflammatory responses following influenza virus or SARS-CoV-2 infection. Our finding that reduction of certain bile acids in the plasma of patients with moderate I/II COVID-19 may provide insight into the variability in clinical disease manifestation in humans and enable approaches for mitigating COVID-19 outcomes.

## Methods

### Ethics statement
All protocols involving specimens from human participants recruited at Juntendo University were reviewed and approved by the Institutional Review of Juntendo University (H20-0222). All animal experiments using mice and hamsters were performed in accordance with University of Tokyo's Regulations for Animal Care and Use, which were approved by the Animal Experiment Committee of the Institute of Medical Science, the University of Tokyo (PA15-92, PA19-87, PA22-33).

### Patients
This is a case-control monocentric study that included 46 patients with COVID-19, hospitalized in the Juntendo University Hospital. All patients were recruited from October 2020 to March 2021. Reverse transcription-polymerase chain reaction (RT-PCR) tests using nasal or pharyngeal swabs were performed for diagnosis of COVID-19. This study was approved by the Ethics Committee of Juntendo University (No. H20-0222) and written informed consent was obtained from all participants. Electronic medical records of the patients' clinical characteristics on admission were collected. Data included self-reported symptoms before PCR testing, the patient's medical history, comorbidities, medical treatment received, blood examination result on admission, and chest x-ray and CT results. Data on COVID-19 management were collected as well. Self-reported symptoms on the day of the PCR test were collected via a questionnaire. All imaging and blood test results were collected on admission.

### Animals
Six-week-old female C57BL/6JJmsSlc mice and 4-week-old female Syrian hamsters obtained from Japan SLC, Inc. were used as WT controls. For some experiments we used aged (52- to 122-week-old) female C57BL/6JJcl mice obtained from CLEA Japan, Inc. (Fig. 1e, f and Supplementary Figs. 7 and 10). B6.Cg-Tg(K18-ACE2)2Prlmn/J (K18-hACE2) mice were purchased from The Jackson Laboratory and subsequently bred at The University of Tokyo. Mice were kept in an incubator (MIR-154, PHCbi, Japan) at 4 °C (80% relative humidity), 22 °C (53% relative humidity), or 36 °C (25% relative humidity). Cold, RT, or high-heat exposures were started 7 days before infection and continued for the entire duration of the experiments. These mice were allowed free access to food and drinking water and kept on a 12 h light/dark cycle.

All animals used in this study were fed with a γ-ray-sterilized normal diet (CE-2, CLEA Japan, Inc.) or AIN93G-fomula diet (Oriental Yeast, Japan) as a LF diet. We confirmed that drinking water did not freeze at 4 °C. All animals were euthanized with deep anesthesia and cervical dislocation. All animal experiments were performed in accordance with The University of Tokyo's Regulations for Animal Care and Use, which were approved by the Animal Experiment Committee of the Institute of Medical Science, The University of Tokyo (PA15-92, PA19-87, PA22-33).

### Antibiotic treatment
Mice were treated with ampicillin (1 g/L; Nacalai Tesque), vancomycin (500 mg/L; Duchefa Biochemie), neomycin sulfate (1 g/L; Nacalai Tesque), metronidazole (1 g/L; Nacalai Tesque), gentamicin (10 mg/L; Nacalai Tesque), 1% penicillin-streptomycin (P/S) (Nacalai Tesque, 09367-34), and 1% amphotericin B (Nacalai Tesque, 02892-54) in drinking water as previously described[11,54]. Antibiotic-containing water was changed twice a week. Antibiotic treatment was started 2 weeks before infection and continued for the entire duration of the experiments.

### Bile acids, GW 4064, or HY-14229 treatment
Mice or hamsters were treated with 0.5 mM of CA (Nacalai Tesque, 08843-72), DCA (Nacalai Tesque, 10711-22), UDCA (AdipoGen Life Sciences, CDX-U0019-G025), TDCA (Nacalai Tesque, 32740-04), or GCA (Toronto Research Chemicals Inc., G641370) in drinking water. For some experiments, mice or hamsters were treated with 0.1 mM of GW 4064 (MyBioSource.Inc., MBS576846) or HY-14229 (MedChem Express, HY-14229) in drinking water. Bile acids, GW 4064, or HY-14229 treatment was stared 7 days before infection and continued for the entire duration of the experiments.

### Cell culture
Madin-Darby canine kidney (MDCK) cells were maintained in minimal essential medium (MEM) (Nacalai Tesque, 21443-15) supplemented with 10% v/v fetal bovine serum (FBS) and 1% v/v penicillin (100 units/ml)/streptomycin (100 μg/ml). VeroE6 cells stably expressing transmembrane protease serine 2 (VeroE6/TMPRSS2; JCRB Cell Bank 1819) were maintained in Dulbecco's modified Eagle's medium (DMEM) (low-glucose) (Nacalai Tesque, 08456-65) supplemented with 10% v/v FBS, 1% v/v penicillin/streptomycin (P/S), and G418 (1 mg ml$^{-1}$; Nacalai Tesque, 16512-94). L929 cells were maintained in DMEM (high-glucose) (Nacalai Tesque, 08458-45) supplemented with 10% FBS and 1% P/S. To prepare bone marrow-derived macrophages, bone marrows from the tibia and femur were obtained by flushing with DMEM. Bone marrow cells were cultured with DMEM supplemented with 10% FBS, L-glutamine, 1% P/S, and 30% L929 supernatant containing the macrophage colony-stimulating factor at 37 °C for 5 days[55].

### Virus infection in vivo
A mouse-adapted influenza A virus strain A/Puerto Rico/8/1934 (PR8) and a human isolate of the 2009 pandemic influenza A virus strain A/Narita/1/2009 (pdm09) were grown in allantoic cavities of 10-d-old fertile chicken egg for 2 days at 35 °C. Viral titers were quantified by a standard plaque assay using MDCK cells and viral stock was stored at −80 °C. For influenza virus infection, mice were infected by intranasal application of 30 μL of virus suspension (500–2000 pfu of influenza virus in PBS) under isoflurane anesthesia. An original SARS-CoV-2 strain bearing aspartic acid at position 614 of spike (S) protein (S-614D) and SARS-CoV-2 B.1.1.7 (alpha) variant were grown in VeroE6/TMPRSS2 cells for 2 days at 37 °C. Viral titers were quantified by a standard plaque assay using VeroE6/TMPRSS2 cells and viral stock was stored at −80 °C. For SARS-CoV-2 infection, K18-hACE2 mice were infected by intranasal application of

50 μL of virus suspension (5 × 10⁴ pfu of SARS-CoV-2 in PBS) under isoflurane anesthesia. Syrian hamsters were infected by intranasal application of 150 μL of virus suspension (1.5 × 10⁶ pfu of SARS-CoV-2 in PBS) under isoflurane anesthesia. ZIKV (ATCC VR-84) was grown in Vero cells for 2 days at 37 °C. Viral titers were quantified by a standard plaque assay using Vero cells and viral stock was stored at −80 °C. For ZIKV infection, mice were injected intravenously with 1.5 × 10⁷ pfu of ZIKV in PBS under isoflurane anesthesia. SFTSV (a gift from S. Morikawa, National Institute of Infectious Diseases, Tokyo, Japan) was grown in Vero cells for 3 days at 37 °C. Viral titers were determined by 50% tissue culture infectious dose (TCID$_{50}$) assay according to published procedures[56] and viral stock was stored at −80 °C. For SFTSV infection, mice were injected intravenously with 5 × 10⁶ TCID$_{50}$ of SFTSV in PBS under isoflurane anesthesia. EMCV was grown in L929 cells for 15 h at 37 °C. Viral titers were quantified by a standard plaque assay using L929 cells and viral stock was stored at −80 °C. For EMCV infection, mice were injected intraperitoneally with 100 pfu of EMCV in PBS under isoflurane anesthesia.

Body temperature was measured by inserting a rodent thermometer (Natsume Seisakusho, KN-91-AD1687) orally. Baseline body weights were measured before infection. Mice and hamsters were deemed to have reached endpoint at 70% of starting weight or after reaching body temperature of 25 °C or lower. Animals were euthanized under anesthesia with an overdose of isoflurane if severe disease symptoms or weight loss were observed. All experiments with SARS-CoV-2 were performed in enhanced biosafety level 3 (BLS-3) containment laboratories at the University of Tokyo, in accordance with the institutional biosafety operating procedures.

### Virus infection in vitro
MDCK cells in 6-well plates were infected with influenza virus PR8 at a multiplicity of infection (MOI) of 0.01 in the presence or absence of bile acids, GW 4064 (a FXR agonist), or HY-14229 (a TGR5 agonist) for 1 h at 37 °C, and cultured with Opti-MEM (Invitrogen, 31985-070) containing acetyltrypsin (10 mg/mL) in the presence or absence of bile acids, GW 4064, or HY-14229 for additional 23 h. VeroE6/TMPRSS2 cells in 6-well plates were infected with SARS-CoV-2 at an MOI of 0.01 in the presence or absence of bile acids, GW 4064, or HY-14229 for 1 h at 37 °C, and cultured with DMEM (low-glucose) supplemented with 5% FBS, 1% P/S, and 1% HEPES buffer (Nacalai Tesque, 17557-94) in the presence or absence of bile acids, GW 4064, or HY-14229 for additional 23 h.

### Western blot analysis
The virus-infected MDCK or VeroE6/TMPRSS2 cells in 6-well plates were washed with PBS and lysed in 500 μl of 1× TNT buffer (50 mM Tris [pH 7.5], 150 mM NaCl, 1% Triton X-100, 1 mM EDTA, 10% glycerol). Lysates were centrifuged at 20,630 × $g$ for 10 min at 4 °C. The supernatant was mixed with 4× lithium dodecyl sulfate (LDS) sample buffer (Invitrogen, NP0007) and 10× sample reducing agent (Invitrogen, NP0009). Samples were boiled for 5 min and fractionated by NuPAGE 10% Bis-Tris Protein Gels (Invitrogen, NP0316BOX) and electroblotted onto polyvinylidene difluoride (PVDF) membranes (Bio-Rad Laboratories, 170-4156). The membranes were incubated with mouse anti-influenza A virus NS1 (Santa Cruz, sc-130568; 1:1000), mouse anti-influenza A virus M2 (Santa Cruz, sc-32238; 1:1000), mouse anti-α tubulin (Santa Cruz, sc-32293; 1:2000), or rabbit anti-SARS-CoV-2 nucleocapsid (Cell Signaling, 33336; 1:1000) antibody, followed by incubation with horseradish peroxidase-conjugated anti-mouse IgG (Jackson Immuno Research Laboratories, 115-035-003; 1:10,000) or anti-rabbit IgG (Invitrogen, G-21234; 1:10,000). The PVDF membranes were then treated with Chemi-Lumi One Super (Nacalai Tesque, 02230-30) to elicit chemiluminescent signals, which were detected and visualized using an LAS-4000 Mini apparatus (GE Healthcare).

## Quantitative PCR

Total RNA was extracted from cells using TRIzol reagent (Invitrogen, 15596018) and reverse transcribed into cDNA using SuperScript III reverse transcriptase (Invitrogen, 18080085) with an oligo (dT) primer. TB Green Premix Ex Taq II (TaKaRa, RR820A) and a LightCycler 1.5 instrument (Roche Diagnostics) were used for quantitative PCR with the following primers: influenza virus NP forward, 5'-agaacatctgacatgaggac-3', and reverse, 5'-gtcaaaggaaggcacgatc-3'.

## ELISA

Cell-free supernatants or lung washes were analyzed for the presence of IFN-α (Hycult Biotech, HM1001; PBL Assay Science, 32100-1), IFN-γ (eBioscience, 14-7311-85 and 13-7312-85,), IL-1β (eBioscience, 14-7012-85 and 13-7112-85), and IL-6 (eBioscience, 14-7061-85 and 13-7062-85) using an enzyme-linked immunosorbent assay (ELISA) utilizing paired antibodies. IFN-β (PBL Assay Science, 42400-1), IFN-λ (PBL Assay Science, 62830-1), or CXCL1 (Proteintech, KE10019) ELISA was performed according to the manufacturer's instructions. Absorbance at 450 nm was measured by using Microplate Manager version 6 (Bio-Rad).

## LDH releases assay

LDH release assays (Promega, G1780) were performed according to the manufacturer's instructions. LDH release data were used to account for cell death. The data are expressed as percentage of maximum LDH release.

## Flow cytometry

The single-cell suspensions of lung samples were prepared as previously described[57]. Briefly, lungs were perfused with 10 ml PBS through the right ventricle, minced using razor blades, and incubated in HBSS containing 2.5 mM Hepes and 1.3 mM EDTA at 37 °C for 30 min. The cells were resuspended in RPMI containing 5% FBS, 1 mM CaCl₂, 1 mM MgCl₂, 2.5 mM Hepes, and 0.5 mg/ml collagenase D (Roche) and incubated at 37 °C for 60 min. A single-cell suspension was prepared after red blood cell lysis. The resulting cells were filtered through a 70-μm cell strainer (BD). For neutrophil staining, cells were incubated with APC-labeled anti-Ly6G (Invitrogen, 17-9668-82; 1:200) and eFluor 450-labeled anti-Ly6C (Invitrogen, 48-5932-82; 1:200) (Supplementary Fig. 37). For the detection of influenza virus-infected cells, MDCK cells were fixed and permeabilized using a Cytofix/Cytoperm kit (BD Biosciences, 554714), and intracellularly stained with FITC-labeled mouse anti-influenza virus NP (abcam, ab20921; 1:100) antibody. Flow cytometric analysis was performed with a FACSVerse flow cytometer (BD Biosciences). The final analysis and graphical output were performed using FlowJo software (Tree Star, Inc.).

## Metabolite extraction and CE-TOFMS measurements

Metabolites in the cecal and serum samples were analyzed as previously described[58,59]. Briefly, cecal and serum metabolites were extracted by vigorous shaking with methanol containing methionine sulfone and D-camphol-10-sulfonic acid (CSA) as the internal standards. Then the mixture was cleaned with chloroform and water extraction. The suspension was centrifuged at $4600 \times g$ for 15 min at 4 °C, and the resulting supernatant was transferred to a 5-kDa-cutoff filter column (Ultrafree MC-PLHCC 250/pk for Metabolome Analysis, Human Metabolome Technologies, Tsuruoka, Yamagata, Japan). The flow-through was dried under vacuum, and the residue then was dissolved in Milli-Q water containing reference compounds (200 μM each of 3-aminopyrrolidine and trimesate). The levels of extracted metabolites were measured in both positive and negative modes by CE-TOFMS as previously described[60]. All CE-TOFMS experiments were performed using an Agilent capillary electrophoresis system and Agilent G3250AA LC/MSD TOF system (Agilent Technologies, Santa Clara, California, U.S.A).

Raw data was analyzed using our proprietary automatic integration software MasterHands ver. 2.19.0.1[60]. Annotation tables were produced from standard compound measurements and aligned with the datasets according to similar m/z values and normalized migration time. Then, peak areas were normalized against the internal standards methionine sulfone and CSA for cationic and anionic metabolites, respectively. Amounts of each metabolite were calculated based on their relative peak areas and the concentrations of the standard compounds. Principal component analysis of CE-TOFMS data was performed using SIMCA 15 (Umetrics, Umea, Sweden). For the analysis, concentrations below the detection limit were substituted with zero, and metabolites whose levels were below the detection limit in all the samples were excluded.

## Quantification of serum and liver bile acids via LC-MS/MS

Serum and liver metabolites were extracted from 50 μL serum or 50 μg liver suspended in 250 μL of 50% methanol in Milli-Q water supplemented with 20 μM CSA as an internal standard. The samples were homogenized using a multi-sample homogenizer (Shake Master Neo, Bio Medical Science) and centrifuged at $20,400 \times g$ for 10 min at 10 °C, and supernatants were harvested. Liquid chromatography was performed using an Agilent 1290 UPLC system (Agilent Technologies) with gradient elution from an ACQUITY UPLC HSS T3 column (1.8 μm, 50 mm × 2.1 mm ID; Waters) maintained at 45 °C with a mobile phase flow rate of 0.3 ml/min[61]. Gradient elution mobile phases comprised A (water, 50 mM ammonium formate) and B (methanol, 50 mM ammonium formate). The gradient was initiated at 10% B, increasing linearly to 50% at 0.1 min, held at 50% B at 2 min, increasing linearly again to 80% at 9.5 min, and then increasing linearly to 100% at 10.5 min; this was maintained until 12 min, with subsequent re equilibration at 10% B for a further 5 min. The sample temperature was maintained at 4 °C in the autosampler prior to analysis. The sample injection volume was 1 μL. Mass spectrometry (MS) analysis was performed using an Agilent 6490 triple quadrupole mass spectrometer MS (Agilent Technologies) equipped with an Agilent Jet Stream ESI probe in negative-ion mode. A capillary voltage of −3500 V, a source temperature of 200 °C, gas flow of 14 L/min, nebulizer of 50 psi, sheath gas temperature of 250 °C, and a sheath gas flow of 11 L/min were used. Optimized m/z values of precursor and product ions, dwell times, collision voltages, and retention times selected for each compound are indicated previously[62]. Concentrations of individual BAs were determined from the peak area in the chromatogram detected with MRM relative to the internal standard, CSA.

## Quantification of cecal organic acids via GC/MS

Organic acid concentrations of cecal contents were determined by gas chromatography–mass spectrometry (GC/MS)[63]. In brief, 10 mg cecal contents were disrupted using 3 mm zirconia/silica beads (BioSpec Products) and homogenized with extraction solution containing 100 μl of internal standard (100 μM crotonic acid), 50 μl HCl and 200 μl ether. After vigorous shaking using Shakemaster neo (Bio Medical Science) at 1500 r.p.m. for 10 min, homogenates were centrifuged at 1000 g for 10 min and then the top ether layer was collected and transferred into new glass vials. Aliquots (80 μl) of the ether extracts were mixed with 16 μl N-tert-butyldimethylsilyl-N-methyltrifluoroacetamide (MTBSTFA). The vials were sealed tightly, heated at 80 °C for 20 min in a water bath, and then left at room temperature for 48 h for derivatization. The derivatized samples were run through a 6890 N Network GC System (Agilent Technologies) equipped with HP-5MS column (0.25 mm × 30 m × 0.25 μm) and 5973 Network Mass Selective Detector (Agilent Technologies). Pure helium (99.9999%) was used a carrier gas and delivered at a flow rate of 1.2 ml per min. The head pressure was set at 97 kPa with split 20:1. The inlet and transfer line temperatures were 250 and 260 °C, respectively. The following temperature program was used: 60 °C (3 min), 60–120 °C (5 °C per

min), 120–300 °C (20 °C per min). One microliter of each sample was injected with a run time of 30 min. Organic acid concentrations were quantified by comparing their peak areas with the standards.

## Fecal DNA extraction

Fecal DNA isolation was performed as described previously, with some modifications[64]. Each freeze-dried fecal sample was combined with four 3.0 mm zirconia beads, ~100 mg of 0.1 mm zirconia/silica beads, 400 μL DNA extraction buffer (TE containing 1% (w/v) sodium dodecyl sulfate), and 400 μL of phenol/chloroform/isoamyl alcohol (25:24:1) and vigorous shaked (1500 rpm for 15 minutes) using a Shake Master (Biomedical Science, Shinjuku, Tokyo, Japan). The resulting emulsion was centrifuged at 17,800 × g for 10 minutes at room temperature, and bacterial genomic DNA was purified from the aqueous phase by a standard phenol/chloroform/isoamyl alcohol protocol[64]. RNA was digested in the sample by RNase A treatment; the resulting DNA sample then was purified again, by another round of phenol/chloroform/isoamyl alcohol treatment.

## 16S rRNA gene sequencing

16S rRNA genes in the fecal microbiota DNA samples were analyzed using a Miseq sequencer (Illumina, San Diego, California, U.S.A.). The V1–V2 region of the 16 S rRNA genes was amplified from the DNA (~10 ng per reaction) using a universal bacterial primer set consisting of primers 27Fmod with an overhang adapter (5′-AGRGTTTGA-TYMTGGCTCAG-3′) and 338R with an overhang adapter (5′- TGC TGCCTCCCGTAGGAGT-3′)[65]. Polymerase chain reaction (PCR) was performed with Tks Gflex DNA Polymerase (Takara Bio, Inc., Kusatsu, Shiga, Japan), and amplification via the following program: one cycle denaturation at 98 °C for 1 min; 20 cycles of amplification at 98 °C for 10 s, 55 °C for 15 s, and 68 °C for 30 s, with final extension at 68 °C for 3 min. The amplified products were purified using Agencourt AMPure XP kits (Beckman Coulter, Atlanta, GA, USA). The purified products were then further amplified using a primer pair as follows: a forward primer (5′-AATGATACGGCGACCA CCGAGATCTACAC-NNNNNNN N-TATGGTAATTGTAGRGTTTGATYMTGGCTCAG-3′) containing the p5 sequence, a unique 8-bp barcode sequence for each sample (indicated by the string of Ns), and an overhang adapter, as well as a reverse primer (5′-CAAGCAGAAGACGGCATACGAGAT-NNNNNNNNAGTCAG TCAGCCTGCTGCCTCCCGTAGGAGT-3′) containing the P7 sequence, a unique 8-bp barcode sequence for each sample (indicated by the string of Ns), and an overhang adapter. After purification using Agencourt AMPure XP kits, the purified products were mixed in approximately equal molar concentrations to generate a 4 nM library pool, after which the final library pool was diluted to 6 pM, including a 10% Phix Control v3 (Illumina, San Diego, California, U.S.A.) spike-in for sequencing. Finally, Miseq sequencing was performed according to the manufacturer's instructions. In this study, 2 × 300-bp paired-end sequencing was employed. The microbiome analysis data have been deposited in the DDBJ Sequence Read Archive (http://trace.ddbj.nig.ac.jp/dra/) under accession number DRA015846.

## Analysis of 16S rRNA gene sequences using QIIME 2

Analysis of 16S rRNA gene sequences was performed as described with some modifications[66]. In brief, filter-passed reads were processed using Quantitative Insights into Microbial Ecology (QIIME) 2 (2019.10.0)[67]. Denoising and trimming of sequences were carried out using DADA2. The first 20 bp and 19 bp were trimmed from the 5′ end of both forward and reverse reads, respectively, to remove primer sequences. The resulting 135-bp and 220-bp reads from the respective 5′ ends were used for subsequent steps. Taxonomy was assigned using the SILVA132 database using the Naive Bayesian Classifier algorithm[68,69]. Alpha diversity of gut microbiota was analyzed using observed species, Chao 1, and Shannon indices. PCoA based on Uni-Frac distances and ANOSIM was carried out using QIIME 2.

## Statistical analysis

Statistical significance was tested using nonparametric one-way analysis of variance (ANOVA) with Tukey's multiple comparison test, nonparametric Mann-Whitney $t$ test, or Student's two-tailed, unpaired $t$ test where indicated in the figure legend, using PRISM software (version 5; GraphPad software). $P < 0.05$ was considered statistically significant.

## Reporting summary

Further information on research design is available in the Nature Portfolio Reporting Summary linked to this article.

## Data availability

The authors declare that the data supporting the findings of this study are available within the article and its Supplementary Information files, or are available on request. Source data are provided with this paper. The microbiome analysis data have been deposited in the DDBJ Sequence Read Archive (http://trace.ddbj.nig.ac.jp/dra/) under accession number DRA015846. The CE-TOFMS-based metabolome data are available at the NIH Common Fund's National Metabolomics Data Repository (NMDR) website, the Metabolomics Workbench, https://www.metabolomicsworkbench.org, where they have been assigned Project ID (ST002476 and ST002479). The data can be accessed directly via the Project https://doi.org/10.21228/M8113P. Source data are provided with this paper.

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

## Acknowledgements

We thank Akiko Iwasaki (Yale University) for critical reading of the manuscript, Yoshihiro Kawaoka (University of Wisconsin and University of Tokyo) for providing SARS-CoV-2/UT-NCGM02/Human/2020/Tokyo, Masayuki Saijo (National Institute of Infectious Diseases) for providing the SARS-CoV-2 QK002 variant. Flow cytometric analysis was performed in the IMSUT FACS Core laboratory. This work was supported in part by research grants from the Japan Agency for Medical Research and Development (JP223fa627001 to T.I.), JSPS KAKENHI (22H03541 to S.F.), AMED-CREST (JP22gm1010009 to S.F.), JST ERATO (JPMJER1902 to S.F.), the Food Science Institute Foundation (to S.F.), the Metagen Therapeutics, Inc. (to S.F. and T.I.), the MSD Life Science Foundation (to T.I.), the Takeda Science Foundation (to T.I.), and the Suzuken Memorial Foundation (to T.I.). M. Moriyama is the Research Fellow of the Japan Society for the Promotion of Science. The metabolomics work is supported by National Institutes of Health grant U2C-DK119886.

## Author contributions

S.F. and T.Ichi. conceived the study and designed the experiments; M.N., M.M., and T.Ichi. performed animal experiments; T.Ishi., A.H., and S.F. conducted metabolome analysis; H.M. and T.Nai. collected clinical samples; C.I., H.W., T.Nak., and T.Y. analyzed data; I.K., D.I., and A.N. provided reagents and advice; and S.F. and T.Ichi. wrote the paper.

## Competing interests

T.N., T.Y., D.I., and S.F. are founders and T.N. and D.I. are board members of Metagen Therapeutics, Inc., a company involved in the microbiome-based medical and drug discovery. T.Y. and S.F. are founders and board members of Metagen, Inc., a microbiome-based healthcare company. The other authors declare no competing interests.
