## [Peer Review File · Nature Communications]

High body temperature increases gut microbiota-dependent host resistance to influenza A virus and SARS-CoV-2 infectionREVIEWER COMMENTS

Reviewer #1 (Remarks to the Author):

In this manuscript, Nagai et al assess the role of heat exposure in resistance to SARS-CoV-2 and influenza viruses, and its relationship to bile acid signaling. Overall, this is a strong manuscript with findings well supported by the data, and employing appropriate methods. Authors should also be commended for validating their findings across a broad range of viral infection systems. However, I have the following concerns.

Major concerns:

1. Supplementary figure 1b: authors should clarify the time of day at which these pictures were taken, and confirm that they are from comparable timepoints, given the major changes in mouse behavior over the light/dark cycle.
2. Figure panels in Figure 3a-d do not match with the in-text description and with figure legend
3. Line 165-167: "the level of propionate and both primary and secondary bile acids were significantly increased in the high heat". This statement is not supported by the data: levels of propionate were not significantly altered in Figure S12.
4. Patient parameters are missing (sex, age, comorbidities).
5. While the findings in COVID-19 patients are intriguing, it is essential that the authors confirm that the observed differences in bile acids between minor and moderate illness are not a result of other differences between these two groups, such as age.
6. Methods details are lacking on how annotations in Supplementary Figure 12b were generated. Authors should also provide a supplemental figure supporting these annotations (for example, demonstrating matching to standards).

Minor concerns:

1. Line 68: a key reference missing is PMID: PMC5555589
2. Figure S13 title does not match legend text (serum vs cecal contents). Same issue for S14 (liver vs serum). Authors must clarify the specific tissue being displayed.
3. Line 206-207: "Several other possible mechanisms could explain how lower concentration of DCA in the serum might increase host resistance to influenza virus infection." should read: "Several other possible mechanisms could explain how higher concentration of DCA."
4. Line 866: define CSA
5. Public deposition of metabolomics data and metadata is recommended

Reviewer #2 (Remarks to the Author):

This is a conceptually interesting paper reporting on several studies that examined multiple timely ideas, including how higher or lower temperature and aging influence antiviral defenses, and what role the microbiome and other host factors play in variable responses to infection. Some of the core observations, such as modulation of host resistance to influenza virus by changing ambient temperature and alterations in microbiota have been previously reported, and this paper reports an extension of these observations. Singly and in combination these are interesting questions. The integrated usage of in vitro and in vivo systems is a major strength of this study. While many of the findings are intriguing, there are numerous aspects of the rationale and reporting that make it difficult to appreciate and evaluate the findings. Addressing some of these issues can likely be accomplished by focusing the paper more clearly, clarifying details throughout the text, particularly in the Results, figure legends and Materials and Methods sections, and by being more circumspect about interpretation of the data and integration with prior studies. However, addressing several of the concerns raised may require additional experimental work, and findings from these follow up experiments may change some of the conclusions.

Rationale

The authors are weaving together efforts to better understand how cooler and higher body temperature (or being in extremely cold or extremely hot environments) affects host resistance to viral

infection, as well as why older adults appear to have more severe clinical outcomes due virus infections, and what role bile acid metabolites and the gut microbiota play in mediating differential pathophysiology to infection. While the importance of understanding how ambient environment, age, and commensal microbiota affect clinical outcomes due virus infection are all important, several aspects of the rationale for the work reported in this paper were insufficient or confusing. A couple examples are provided to try to illustrate this concern and suggestion to help others better understand this work.

- Throughout the paper there seems to be a presumed equivalence of elevating the temperature in the ambient environment with fever. Fever is a biological process that includes, in some animals, a higher temperature (although during acute virus infection mice experience diminished body temperature, which the authors observed in the control (22°C) mice). Moreover, fever is accompanied by the production of acute phase response and a network of inflammatory mediators. It is not clear that being a hotter ambient temperature triggers the same pathophysiological and immunological events as fever.
- The rationale for including mice at excessively cold temperature is not explained. If there is an intentional health relevance to this experimental paradigm, then the authors need to provide it. Related to this, the possible deleterious impact of maintaining mice at extreme temperatures for long periods of time was not adequately considered. In particular, all of the mice kept at 4°C died quite rapidly after infection. These consequences could be due to stress or other factors of maintaining mice in stressful conditions, and not be related to any specific impact of low temperature on virus infection or the immune response to the infection.

Results and Data interpretation

This manuscript presents several novel and very interesting observations; yet, some of the data do not robustly or directly support the conclusions, and additional experimental work is needed. Related to this, there are instances in which there is not sufficient thoroughness or clarity of the work performed. Several examples are provided to help convey this concern.

- The Results section opens recapitulating some key observations that have been previously reported, using a mixture of supplementary figures and panels in Figure 1. These data provide a helpful foundation for the paper as a whole, although the authors may wish to consider leading off with new observations or a single key prior observation that drives the work in this new paper forward, rather than opening with extensive descriptions of data that have been previously shown. Related to this, within this section, there are many details that were not well explained, such as the rationale for partial hepatectomy, and superficial mention of many other experiments, which are in 8 supplementary figures.
- The narrow focus on bile acids and TGR5 axis was not strongly substantiated within the main body of the Results section. There are 4 supplementary figures included, and these seem to be provided to help readers make this leap. However, it was cumbersome to evaluate how these supplemental materials lead directly to deoxycholic acid and TGR5 axis. Also, within the work presented in Fig 3, additional details are needed, including (but not limited to) the administered dose, route of exposure and evidence of efficacy of HY-14229 given to mice.
- The findings presented in Figure 4 were also little bit confusing for similar reasons. Also, I appreciate that the authors included a dose-response assessment of deoxycholic acid treatment of cells in vitro (Fig 4g), but how this informed the in vivo administration of mice with deoxycholic acid was not clear. To evaluate the data, considerably more information is needed. I also wondered whether it is possible that the lower levels of CXCL1 and fewer neutrophils are a consequence of reduced lung viral load, as opposed to being due to factors derived from the gut (or other) microbiota. This should probably be considered.
- The paper includes some very limited findings based on 46 patients with COVID-19. Calling this a 'case-control study' does not seem accurate. All of the patients had COVID-19. The more substantive concern is that duration of illness, age, and many other possible explanations for differences observed (e.g Fig 5f and 5i) were not considered adequately. Also, no information on these subjects is provided, including exclusion or inclusion criteria, and other population characteristics, etc.

Reporting

There is a need for key details to be provided, particularly but not limited to in the Methods and figure

legends. In addition to concerns noted above, a few additional examples are provided to better illustrate this concern:

- The authors indicate that they used age and sex matched mice, but do not indicate whether male and female mice were used, or if all data are from one sex. Likewise, the age(s) of the mice are difficult to appreciate.
- Some of the figure legends don't seem to align with what is presented in the Figure (e.g., Fig 3).
- The number of mice per group, and for in vitro studies the number of independent replicates, is not indicated.
- Although the Methods has a brief summary of statistical analysis, it would be helpful to indicate in each figure legend which statistical test(s) was(were) used.
- Doubtless a minor wording issues, but there seem to be some descriptive errors that heighten the sense that more experimental work or more careful attention to detail is needed (e.g., for the results presented figure 4, the text indicates that they isolated lymphocytes, yet the data are about neutrophils, which are not lymphocytes).
- The authors may want to consider more judicious use of abbreviations as this would enhance overall clarity of their paper.

Focus and organization

- This paper describes some intriguing observations. Yet, overall, these findings and the paper as a whole is weakened rather than enhanced by the attempt to include too many things. As an example, in a follow up to the findings in Figure 2 (exploring whether gut microbiota or microbial metabolites are needed to increase host resistance to influenza virus in heat exposed mice), the authors infected heat-treated mice that were on a low fiber diet or that were treated with antibiotics. In both instances, the mice were not able to survive infection. These findings suggest that alterations in gut microbiome could underly this effect of high heat, but this experiment did not include key controls (e.g., mice at 22°C, and put on low fiber diet or treated with antibiotics), nor does this single experiment solidly establish a causal relationship between gut microbiota and the consequences of being maintained in a 36°C environment. Further experimental work is needed. As an example, one could transfer intestinal contents from one group of mice (e.g., those maintained at 36°C) to control mice and vice versa further test causality. There are additional approaches one could take, and I hesitate to be overly proscriptive in which experiments to prioritize.
- The authors are encouraged to restructure the paper with a focused hypothesis or related set of hypotheses, and experimental work that relates directly to testing this hypothesis. This may necessitate additional experiments to more rigorously test the hypothesis(es) and support the conclusion(s). Related to this suggestion, the Results section as a whole relies very heavily on directing readers to extensive supplementary materials. This is cumbersome, and many of these findings were not presented in sufficient detail or experimental depth.

Reviewer #3 (Remarks to the Author):

In this manuscript, Nagai et al explore the impact of body temperature on host resistance or susceptibility to a variety of viruses with the primary focus on influenza and SARS-CoV-2. By exposing mice (or hamsters) to different temperatures prior to- and during infection, they demonstrated that exposure to higher temperatures leads to increased resistance to infection as measured by morbidity and viral titers. The authors demonstrate that high temperatures results in elevated gut microbial-expressed bile acids that are able to reduce viral replication in vitro and in vivo. These are interesting and significant studies that will generate discussion in the field with the appropriate modifications.

dSpecific Comments:

1. More detail is required to review the rigor of the studies. This includes the number of animals and replicates used per study, the anesthesia used for viral inoculation, and more information on temperature modulation and core body temperature measurements.
2. A critical component of influenza infection and transmission is relative humidity. Please discuss if relative humidity differed along with temperature.

3. Please discuss how changes in temperature impacted eating habits and respiration rates and whether the respiration rates could impact the initial viral inoculation.
4. Are similar results obtained if you do tracheal inoculation.
5. Higher temperature also protected against SARS-CoV-2 yet there was no difference in viral titers. This leads to questions about the mechanism of action. Please discuss.
6. Figure 3 does not align with the text. Please correct.
7. The author's conclude that viral titers are decreased in the lungs of bile acid-treated mice (Fig 3), yet this isn't supported by the data. Were there differences in viral spread within the lungs?
8. The low fiber and antibiotic studies are interesting. Please provide data on the microbiome composition before and at the time of challenge to support your conclusions.
9. High body temperature failed to protect CA09 H1N1 infected mice as well as PR8 infected animals. This leads to questions about the significance of the studies beyond a particular influenza strain.
10. Please describe the limitations of the studies and how this work could translate to humans.

REVIEWER COMMENTS

Reviewer #1 (Remarks to the Author):

In this manuscript, Nagai et al assess the role of heat exposure in resistance to SARS-CoV-2 and influenza viruses, and its relationship to bile acid signaling. Overall, this is a strong manuscript with findings well supported by the data, and employing appropriate methods. Authors should also be commended for validating their findings across a broad range of viral infection systems. However, I have the following concerns.

Major concerns:

1. Supplementary figure 1b: authors should clarify the time of day at which these pictures were taken, and confirm that they are from comparable timepoints, given the major changes in mouse behavior over the light/dark cycle.

We thank the reviewer for pointing this out. As suggested by the reviewer, we have now added the time of day at which these pictures were taken and modified the figure legends for Supplementary Figure 1b to clarify this point.

2. Figure panels in Figure 3a-d do not match with the in-text description and with figure legend

We apologize for this oversight. We have now modified the text (lines 221-223, page 8) and figure legend (lines 552-557, page 20).

3. Line 165-167: “the level of propionate and both primary and secondary bile acids were significantly increased in the high heat”. This statement is not supported by the data: levels of propionate were not significantly altered in Figure S12.

We apologize for this oversight. We deleted the propionate from the sentence (line 204, page 8).

4. Patient parameters are missing (sex, age, comorbidities).

We thank the reviewer for pointing this out. As suggested by the reviewer, we have now added the information about patients' sex, age, and comorbidities in Supplementary Table 1.

5. While the findings in COVID-19 patients are intriguing, it is essential that the authors confirm that the observed differences in bile acids between minor and moderate illness are not a result of other differences between these two groups, such as age.

We thank the reviewer for pointing this out. As suggested by the reviewer, we have now examined the correlation between the levels of plasma bile acids in COVID-19 patients and their age. However, the levels of plasma bile acids in COVID-19 patients were not significantly correlated with the patients' age (see Supplementary Fig. 34).

6. Methods details are lacking on how annotations in Supplementary Figure 12b were generated. Authors should also provide a supplemental figure supporting these annotations (for example, demonstrating matching to standards).

We thank the reviewer for pointing this out. As suggested by the reviewer, we have now added the explanation to clarify this point in the Material and Methods section (lines 1025-1031, page 36).

Minor concerns:

1. Line 68: a key reference missing is PMID: PMC5555589

We apologize for this oversight. We have now added the key reference (Reference #16).

2. Figure S13 title does not match legend text (serum vs cecal contents). Same issue for S14 (liver vs serum). Authors must clarify the specific tissue being displayed.

We apologize for this oversight. We have now modified the figure legends (Supplementary Figs. 15 and 16).

3. Line 206-207: "Several other possible mechanisms could explain how lower concentration of DCA in the serum might increase host resistance to influenza virus infection." should read: "Several other possible mechanisms could explain how higher concentration of DCA."

We apologize for this oversight. We have now replaced "lower" with "higher" (line 247, page 9)

4. Line 866: define CSA

We thank the reviewer for pointing this out. As suggested by the reviewer, we have now defined CSA (line 1013, page 35).

5. Public deposition of metabolomics data and metadata is recommended

We thank the reviewer for pointing this out. As suggested by the reviewer, we have deposited the CE-TOFMS-based metabolome data at Metabolome Workbench under accession <http://dx.doi.org/10.21228/M8113P>. We have now included this explanation to clarify this point in the Material and Methods section (lines 1035-1040, page 36).

Reviewer #2 (Remarks to the Author):

This is a conceptually interesting paper reporting on several studies that examined multiple timely ideas, including how higher or lower temperature and aging influence antiviral defenses, and what role the microbiome and other host factors play in variable responses to infection. Some of the core observations, such as modulation of host resistance to influenza virus by changing ambient temperature and alterations in microbiota have been previously reported, and this paper reports an extension of these observations. Singly and in combination these are interesting questions. The integrated usage of in vitro and in vivo systems is a major strength of this study. While many of the findings are intriguing, there are numerous aspects of the rationale and reporting that make it difficult to appreciate and evaluate the findings. Addressing some of these issues can likely be accomplished by focusing the paper more clearly, clarifying details throughout the text, particularly in the Results, figure legends and Materials and Methods sections, and by being more circumspect about interpretation of the data and integration with prior studies. However, addressing several of the concerns raised may require additional experimental work, and findings from these follow up experiments may change some of the conclusions.

Rationale

The authors are weaving together efforts to better understand how cooler and higher body temperature (or being in extremely cold or extremely hot environments) affects host resistance to viral infection, as well as why older adults appear to have more severe clinical outcomes due virus infections, and what role bile acid metabolites and the gut microbiota play in mediating differential pathophysiology to infection. While the importance of understanding how ambient environment, age, and commensal microbiota affect clinical outcomes due virus infection are all important, several aspects of the rationale for the work reported in this paper were insufficient or confusing. A couple examples are provided to try to illustrate this concern and suggestion to help others better understand this work.

- Throughout the paper there seems to be a presumed equivalence of elevating the temperature in the ambient environment with fever. Fever is a biological

process that includes, in some animals, a higher temperature (although during acute virus infection mice experience diminished body temperature, which the authors observed in the control (22°C) mice). Moreover, fever is accompanied by the production of acute phase response and a network of inflammatory mediators. It is not clear that being a hotter ambient temperature triggers the same pathophysiological and immunological events as fever.

We thank the reviewer for pointing this out. We fully agree the reviewer's comment that it is not clear whether a hotter ambient temperature triggers the same pathophysiological and immunological events as fever. Thus, we have now described the limitations of the studies and explained necessity of future studies to dissect other mechanisms by which high body temperature confers host resistance to viral infection (lines 447-450, page 15, lines 461-463, page 16).

- The rationale for including mice at excessively cold temperature is not explained. If there is an intentional health relevance to this experimental paradigm, then the authors need to provide it. Related to this, the possible deleterious impact of maintaining mice at extreme temperatures for long periods of time was not adequately considered. In particular, all of the mice kept at 4°C died quite rapidly after infection. These consequences could be due to stress or other factors of maintaining mice in stressful conditions, and not be related to any specific impact of low temperature on virus infection or the immune response to the infection.

We thank the reviewer for pointing this out. According to the reviewer's suggestion, we have now added sentences to explain the rationale of this study (lines 81-83, page 3). As for the possible deleterious impact, we have previously showed that cold or high-heat exposure of naïve mice was generally well tolerated (Moriyama et al. PNAS 2019). We have now added a sentence to avoid this confusion (lines 101-102, page 5). In addition, we previously demonstrated that the high heat-exposed mice decreased their food intake and increased autophagy in lung tissue (Moriyama et al. PNAS 2019). We have now added a sentence to avoid this confusion (lines 447-448, page 15).

Results and Data interpretation

This manuscript presents several novel and very interesting observations; yet, some of the data do not robustly or directly support the conclusions, and

additional experimental work is needed. Related this, there are instances in which there is not sufficient thoroughness or clarity of the work performed. Several examples are provided to help convey this concern.

- The Results section opens recapitulating some key observations that have been previously reported, using a mixture of supplementary figures and panels in Figure 1. These data provide a helpful foundation for the paper as a whole, although the authors may wish to consider leading off with new observations or a single key prior observation that drives the work in this new paper forward, rather than opening with extensive descriptions of data that have been previously shown. Related to this, within this section, there are many details that were not well explained, such as the rationale for partial hepatectomy, and superficial mention of many other experiments, which are in 8 supplementary figures.

We thank the reviewer for pointing this out. According to the reviewer's suggestion, we have now added sentences to explain the rationale for partial hepatectomy (lines 145-148, page 6). As for 8 supplementary figures, we have now added sentences to explain rationale for these experiments and data interpretation in the text (lines 120-138, pages 5-6).

- The narrow focus on bile acids and TGR5 axis was not strongly substantiated within the main body of the Results section. There are 4 supplementary figures included, and these seem to be provided to help readers make this leap. However, it was cumbersome to evaluate how these supplemental materials lead directly to deoxycholic acid and TGR5 axis. Also, within the work presented in Fig 3, additional details are needed, including (but not limited to) the administered dose, route of exposure and evidence of efficacy of HY-14229 given to mice.

We thank the reviewer for pointing this out. As suggested by the reviewer, we have now added the administered dose and route of exposure of bile acids, GW 4064 (a FXR agonist), and HY-14229 (a TGR5 agonist) in figure legends (line 563, page 20, line 614, page 23) and the Materials and Methods section (lines 888-896, page 32).

As for evidence of efficacy of HY-14229 in mice, we first examined whether triamterene, an inhibitor of the TGR5 receptor, cancels the effect of HY-14229 to inhibit the virus protein synthesis in vitro. However, the triamterene

did not cancel the effect of HY-14229 to inhibit the virus protein synthesis in vitro (see Figure 1 for reviewer). Thus, we wish to resolve these issues in the future.

Figure 1 for reviewer. MDCK cells were infected with influenza virus A/PR8 in the presence or absence of HY-14229 (1 μ M) and various amounts (1 μ M, 100nM, 10nM, 1nM) of triamterene. Cell lysates were collected at 24 h p.i. and analyzed by immunoblotting with indicated antibodies.

- The findings presented in Figure 4 were also little bit confusing for similar reasons. Also, I appreciate that the authors included a dose-response assessment of deoxycholic acid treatment of cells in vitro (Fig 4g), but how this informed the in vivo administration of mice with deoxycholic acid was not clear. To evaluate the data, considerably more information is needed. I also wondered whether it is possible that the lower levels of CXCL1 and fewer neutrophils are a consequence of reduced lung viral load, as opposed to being due to factors derived from the gut (or other) microbiota. This should probably be considered. We thank the reviewer for pointing this out. We fully agree the reviewer's comment that cytotoxic assay in vitro is not enough to determine the administration dose of bile acids in vivo. Thus, we have now described the limitations of the studies and explained necessity of future studies to determine the administration dose and route to maximize the protective effects of bile acids or the agonists against influenza virus or SARS-CoV-2 infection in vivo (lines 456-461, pages 15-16).

As for the lower levels of CXCL1 and fewer neutrophil recruitments, we showed that the levels of CXCL1 and neutrophil recruitment were significantly

elevated in the lung of high heat-exposed LF-fed and Abx-treated mice compared with high heat-exposed control group (Supplementary Fig. 25), without affecting the virus titer at 2 and 3 d p.i. (Fig. 2g). Thus, these data ruled out this possibility that the lower levels of CXCL1 and fewer neutrophils are a consequence of reduced lung viral load.

- The paper includes some very limited findings based on 46 patients with COVID-19. Calling this a 'case-control study' does not seem accurate. All of the patients had COVID-19. The more substantive concern is that duration of illness, age, and many other possible explanations for differences observed (e.g. Fig 5f and 5i) were not considered adequately. Also, no information on these subjects is provided, including exclusion or inclusion criteria, and other population characteristics, etc.

We thank the reviewer for pointing this out. As suggested by the reviewer, we have now added the information about patients' sex, age, comorbidities, and duration of illness in Supplementary Table 1. In addition, we examined the correlation between the levels of plasma bile acids in COVID-19 patients and their age. However, the levels of plasma bile acids in COVID-19 patients were not significantly correlated with the patients' age (see Supplementary Fig. 34).

Reporting

There is a need for key details to be provided, particularly but not limited to in the Methods and figure legends. In addition to concerns noted above, a few additional examples are provided to better illustrate this concern:

- The authors indicate that they used age and sex matched mice, but do not indicate whether male and female mice were used, or if all data are from one sex. Likewise, the age(s) of the mice are difficult to appreciate.

We thank the reviewer for pointing this out. In the current study, we used 6-week-old female C57BL/6J mice and 4-week-old female Syrian hamsters. For some experiments we used aged (52- to 122-week-old) female C57BL/6J mice (Fig. 1e, f and Supplementary Fig. 7 and 10). We have now included this explanation to clarify this point in the Material and Methods section (lines 864-867, page 31).

- Some of the figure legends don't seem to align with what is presented in the Figure (e.g., Fig 3).

We apologize for this oversight. We have now modified the text (lines 221-223, page 8) and figure legend (lines 552-557, page 20).

- The number of mice per group, and for in vitro studies the number of independent replicates, is not indicated.

We thank the reviewer for pointing this out. As suggested by the reviewer, we have now added the number of animals and replicates used per study (see figure legends).

- Although the Methods has a brief summary of statistical analysis, it would be helpful to indicate in each figure legend which statistical test(s) was(were) used.

We thank the reviewer for pointing this out. As suggested by the reviewer, we have now indicated the statistical test we used (see figure legends).

- Doubtless a minor wording issues, but there seem to be some descriptive errors that heighten the sense that more experimental work or more careful attention to detail is needed (e.g., for the results presented figure 4, the text indicates that they isolated lymphocytes, yet the data are about neutrophils, which are not lymphocytes).

We thank the reviewer for pointing this out. As suggested by the reviewer, we have now replaced "lymphocytes" with "leucocytes" (see figure legends of Fig. 4c, d, h and Supplementary Fig. 25).

- The authors may want to consider more judicious use of abbreviations as this would enhance overall clarity of their paper.

We thank the reviewer for pointing this out. As suggested by the reviewer, we have now deleted some abbreviations such as RH (relative humidity), RT (room temperature) and BMM (bone marrow-derived macrophages) from the main text and Supplementary Figure legends.

Focus and organization

- This paper describes some intriguing observations. Yet, overall, these findings and the paper as a whole is weakened rather than enhanced by the attempt to include too many things. As an example, in a follow up to the findings in Figure

2 (exploring whether gut microbiota or microbial metabolites are needed to increase host resistance to influenza virus in heat exposed mice), the authors infected heat-treated mice that were on a low fiber diet or that were treated with antibiotics. In both instances, the mice were not able to survive infection. These findings suggest that alterations in gut microbiome could underly this effect of high heat, but this experiment did not include key controls (e.g., mice at 22°C, and put on low fiber diet or treated with antibiotics), nor does this single experiment solidly establish a causal relationship between gut microbiota and the consequences of being maintained in a 36°C environment. Further experimental work is needed. As an example, one could transfer intestinal contents from one group of mice (e.g., those maintained at 36°C) to control mice and vice versa further test causality. There are additional approaches one could take, and I hesitate to be overly proscriptive in which experiments to prioritize.

We thank the reviewer for helpful suggestion. Since we focused on the mechanisms by which high heat-exposure of mice increases host resistance to lethal influenza virus infection, we infected high heat-exposed LF-fed or Abx-treated mice with influenza virus. In addition, we found that both primary and secondary bile acids were significantly reduced in the serum of high heat-exposed LF-fed or Abx-treated mice compared with high heat-exposed control mice (Supplementary Fig. 17). These observations led us to focus on the role of bile acids in resistance to influenza virus infection at 22°C. Further, we examined the amounts of DNA isolated from cecal contents and the bacterial clusters in the cecum of high heat-exposed control, LF-fed, and Abx-treated mice. We found that high heat-exposed LF-fed and Abx-treated mice significantly reduced the amounts of DNA isolated from cecal contents compared with high heat-exposed control group (see Supplementary Fig. 12a). In addition, we found that high heat-exposed LF-fed and Abx-treated mice changed the bacterial clusters in the cecum compared with high heat-exposed control group (see Supplementary Fig. 12b-e). These data considerably strengthen our original claim that high heat-exposed mice may increase gut microbial biotransforming reactions to produce the secondary bile acids.

- The authors are encouraged to restructure the paper with a focused hypothesis or related set of hypotheses, and experimental work that relates directly to testing this hypothesis. This may necessitate additional experiments

to more rigorously test the hypothesis(es) and support the conclusion(s). Related to this suggestion, the Results section as a whole relies very heavily on directing readers to extensive supplementary materials. This is cumbersome, and many of these findings were not presented in sufficient detail or experimental depth.

We thank the reviewer for pointing this out. Because we believe that all supplementary materials are necessary to support our hypotheses and conclusions, we felt reluctant to exclude the supplementary data in our current manuscript.

Reviewer #3 (Remarks to the Author):

In this manuscript, Nagai et al explore the impact of body temperature on host resistance or susceptibility to a variety of viruses with the primary focus on influenza and SARS-CoV-2. By exposing mice (or hamsters) to different temperatures prior to- and during infection, they demonstrated that exposure to higher temperatures leads to increased resistance to infection as measured by morbidity and viral titers. The authors demonstrate that high temperatures results in elevated gut microbial-expressed bile acids that are able to reduce viral replication in vitro and in vivo. These are interesting and significant studies that will generate discussion in the field with the appropriate modifications.

Specific Comments:

1. More detail is required to review the rigor of the studies. This includes the number of animals and replicates used per study, the anesthesia used for viral inoculation, and more information on temperature modulation and core body temperature measurements.

We thank the reviewer for pointing this out. As suggested by the reviewer, we have now added the number of animals and replicates used per study (see figure legends), the anesthesia for viral inoculation (lines 925-938, page 33), and information on temperature modulation (lines 869-872, page 31) and body temperature measurements (lines 939-940, page 33).

2. A critical component of influenza infection and transmission is relative humidity. Please discuss if relative humidity differed along with temperature.

We thank the reviewer for pointing this out. According to the reviewer's suggestion, we have now discussed the effects of relative humidity and limitations of the studies (lines 441-463, pages 15-16).

3. Please discuss how changes in temperature impacted eating habits and respiration rates and whether the respiration rates could impact the initial viral inoculation.

We thank the reviewer for pointing this out. According to the reviewer's suggestion, we have now discussed the effects of outside temperature in food intake or respiration rates (lines 447-456, page 15).

4. Are similar results obtained if you do tracheal inoculation ?

We thank the reviewer for helpful suggestion. According to the reviewer's suggestion, we have now examined body weight changes and survival of cold-, RT-, and high heat-exposed mice following intratracheal influenza virus infection (see new Supplementary Fig. 3).

5. Higher temperature also protected against SARS-CoV-2 yet there was no difference in viral titers. This leads to questions about the mechanism of action. Please discuss.

We thank the reviewer for pointing this out. We fully agree the reviewer's comment that bile acids or their receptors signaling protected hamsters from lethal SARS-CoV-2 infection without affecting the virus titers (Supplementary Fig. 27b, Supplementary Fig. 30, and Supplementary Fig.36). According to the reviewer's suggestion, we have now discussed the mechanisms by which high body temperature confers host resistance to viral infections (lines 428-440, page 15).

6. Figure 3 does not align with the text. Please correct.

We apologize for this oversight. We have now modified the text (lines 221-223, page 8) and figure legend (lines 552-557, page 20).

7. The authors conclude that viral titers are decreased in the lungs of bile acid-treated mice (Fig 3), yet this isn't supported by the data. Were there differences in viral spread within the lungs?

We thank the reviewer for pointing this out. We fully agree the reviewer's comment that the virus titers remained similar in water-fed and CA-treated mice at 3 and 6 d p.i. (Fig. 3c). According to the reviewer's suggestion, we have now toned down and modified our original manuscript (lines 227-233, pages 8).

8. The low fiber and antibiotic studies are interesting. Please provide data on the microbiome composition before and at the time of challenge to support your conclusions.

We thank the reviewer for pointing this out. As suggested by the reviewer, we have now examined the amounts of DNA isolated from cecal contents and the bacterial clusters in the cecum of high heat-exposed control, LF-fed, and Abx-treated mice. We found that high heat-exposed LF-fed and Abx-treated mice

significantly reduced the amounts of DNA isolated from cecal contents compared with high heat-exposed control group (see Supplementary Fig. 12a). In addition, we found that high heat-exposed LF-fed and Abx-treated mice changed the bacterial clusters in the cecum compared with high heat-exposed control group (see Supplementary Fig. 12b-e). We have now included these new data in Supplementary Figure 12.

9. High body temperature failed to protect CA09 H1N1 infected mice as well as PR8 infected animals. This leads to questions about the significance of the studies beyond a particular influenza strain.

We thank the reviewer for pointing this out. In original Supplementary Fig. 3 (revised Supplementary Fig. 4), we infected mice with a high dose (3×10^4 pfu) of a human isolate of the 2009 pandemic influenza A virus strain A/Narita/1/2009 (pdm09). Thus, we have now examined body weight changes and survival rate when cold-, RT-, and high heat-exposed mice are challenged with lower doses (3,000 or 500 pfu) of the pdm09 strain. We found that protective effect of high body temperature against the pdm09 strain depends on the viral load (see new Supplementary Fig. 4). These data are consistent with our original data using influenza virus PR8 strain (Fig. 1b, c and Supplementary Fig. 2). Thus, we have now added a sentence to avoid this confusion (lines 118-119, page 5).

10. Please describe the limitations of the studies and how this work could translate to humans.

We thank the reviewer for pointing this out. According to the reviewer's suggestion, we have now described the limitations of the studies and explained necessity of future studies to dissect other mechanisms by which high body temperature confers host resistance to viral infection (lines 441-463, pages 15-16).

REVIEWERS' COMMENTS

Reviewer #1 (Remarks to the Author):

All of my comments have been satisfactorily addressed.

Reviewer #2

These comments were mediated by reviewer #3 and considered to be addressed.

I do think the author's addressed the reviewers comments. My only hesitation is whether stating something as a limitation of a study suffices when the reviewer asked for further experiments. See bullet 2 under "Results and Data interpretation". I would accept it but wanted to bring it to your attention.

Reviewer #3 (Remarks to the Author):

The authors were responsive to previous reviewer's comments. No further concerns.

REVIEWER COMMENTS

Reviewer #2

These comments were mediated by reviewer #3 and considered to be addressed.

I do think the author's addressed the reviewers comments. My only hesitation is whether stating something as a limitation of a study suffices when the reviewer asked for further experiments. See bullet 2 under "Results and Data interpretation". I would accept it but wanted to bring it to your attention.

We thank the reviewer for pointing this out. We would like to consider this in our future work.